



# Evaluating Arctic clouds modelled with the Unified Model and Integrated Forecasting System

Gillian Young[1], Jutta Vüllers[1], Peggy Achtert[2], Paul Field[1,3], Jonathan J. Day[4], Richard Forbes[4], Ruth Price[1], Ewan O'Connor[5], Michael Tjernström[6], John Prytherch[6], Ryan Neely III[1,7], and Ian M. Brooks[1]

[1]Institute for Climate and Atmospheric Science, School of Earth and Environment, University of Leeds, Leeds, UK

[2]Meteorological Observatory Hohenpeißenberg, German Weather Service, Germany

[3]Met Office, Exeter, UK

[4]European Centre for Medium-Range Weather Forecasts, Reading, UK

[5]Finnish Meteorological Institute, Helsinki, Finland

[6]Department of Meteorology, Stockholm University, Sweden

[7]National Centre for Atmospheric Science, School of Earth and Environment, University of Leeds, Leeds, UK

*Correspondence to*: Gillian Young (G.Young1@leeds.ac.uk)

**Abstract:** By synthesising remote-sensing measurements made in the central Arctic into a model-gridded Cloudnet cloud product, we evaluate how well the Met Office Unified Model (UM) and European Centre for Medium-Range Weather Forecasting Integrated Forecasting System (IFS) capture Arctic clouds and their associated interactions with the surface energy balance and the thermodynamic structure of the lower troposphere. This evaluation was conducted using a four-week observation period from the Arctic Ocean 2018 expedition, where the transition from sea ice melting to freezing conditions was measured. Three different cloud schemes were tested within a nested limited area model (LAM) configuration of the UM – two regionally-operational single-moment schemes (UM_RA2M and UM_RA2T), and one novel double-moment scheme (UM_CASIM-100) – while one global simulation was conducted with the IFS, utilising its default cloud scheme (ECMWF_IFS).

Consistent weaknesses were identified across both models, with both the UM and IFS overestimating cloud occurrence below 3 km. This overestimation was also consistent across the three cloud configurations used within the UM framework, with >90% mean cloud occurrence simulated between 0.15 and 1 km in all model simulations. However, the cloud microphysical structure, on average, was modelled reasonably well in each simulation, with the cloud liquid water content ($LWC$) and ice water content ($IWC$) comparing well with observations over much of the vertical profile. The key microphysical discrepancy between the models and observations was in the $LWC$ between 1 and 3 km, where most simulations (all except UM_RA2T) overestimated the observed $LWC$.

Despite this reasonable performance in cloud physical structure, both models failed to adequately capture cloud-free episodes: this consistency in cloud cover likely contributes to the ever-present near-surface temperature bias simulated in every simulation. Both models also consistently exhibited temperature and moisture biases below 3 km, with particularly strong cold biases coinciding with the overabundant modelled cloud layers. These biases are likely due to too much cloud top radiative





cooling from these persistent modelled cloud layers and were interestingly consistent across the three UM configurations tested,
despite differences in their parameterisations of cloud on a sub-grid-scale. Alarmingly, our findings suggest that these biases
in the regional model were inherited from the driving model, thus triggering too much cloud formation within the lower
troposphere. Using representative cloud condensation nuclei concentrations in our double-moment UM configuration, while
improving cloud microphysical structure, does little to alleviate these biases; therefore, no matter how comprehensive we make
the cloud physics in the nested LAM configuration used here, its cloud and thermodynamic structure will continue to be
overwhelmingly biased by the meteorological conditions of its driving model.
**1        Introduction**
The Arctic is warming at more than twice the global average rate (**Serreze and Barry, 2011**; **Cohen et al., 2014**), with recent
evidence suggesting the rate of warming could be up to three times the global average (**AMAP 2021**). Coupled general
circulation models (GCMs) fail to agree on the magnitude of recent warming and exhibit large biases in surface temperature
and energy balance (**Boeke and Taylor, 2016**) driven largely by model parameter uncertainties on a decadal scale (**Hodson et**
**al., 2013**). Biases in such surface properties are also present in atmosphere-only versions of these models with fixed ocean and
sea ice boundaries, indicating that there is an important atmospheric source of disparity between models and reality (**Bourassa**
**et al., 2013**). Arctic clouds have a net warming effect at the surface (**Boucher et al., 2013**) and are likely a contributing factor
to the spread of surface energy balance estimates obtained from these models, with a large spread in cloud fractions, liquid
water paths ($LWP$), and ice water paths ($IWP$) identified in past phases of the Coupled Model Intercomparison Project (CMIP;
**Karlsson and Svensson, 2011**; **Boeke and Taylor, 2016**). Early results from the most recent CMIP indicate that high latitude
discrepancies in cloud fraction are still prevalent in recent revisions of these models (**Vignesh et al., 2020**).
With accelerating Arctic warming, we need to build suitable numerical models to confidently predict how the atmosphere will
change both on short weather prediction and longer climate time scales (**Jung et al., 2016**). Models such as the Met Office
Unified Model (UM) and European Centre for Medium-Range Weather Forecasting (ECMWF) Integrated Forecasting System
(IFS) are commonly used for assessing future Arctic change; however, recent work has shown that, like other large-scale
models, both exhibit surface energy balance discrepancies with comparison to high Arctic observations. In both the UM and
the IFS, these biases have largely been attributed to incorrect cloud cover (**Birch et al., 2012**; **Sotiropoulou et al., 2016**;
**Tjernström et al., 2021**).
Several studies have considered why such large-scale models fail to reproduce observed cloud cover in the high Arctic.
Observations have shown that during summer, Arctic clouds experience episodes of extremely low concentrations of cloud
condensation nuclei (CCN; < 10 cm$^{-3}$) approximately 10—30% of the time (**Mauritsen et al., 2011**; **Tjernström et al., 2014**),
highlighting that model capability to reproduce cloud free conditions in the Arctic is likely dependent upon representing these
low CCN numbers (**Birch et al., 2012**; **Stevens et al., 2017**; **Hines and Bromwich, 2017**). Such conditions are difficult to
simulate with large-scale numerical models utilising single-moment microphysics schemes with assumed constant droplet
number concentrations, $N_d$. Both the IFS and the UM make such assumptions in their current global operational configurations:
while climatological aerosol concentrations are referenced in the calculations of the first and second indirect effect, droplet
number cannot evolve independently of cloud liquid mass.





The operational single-moment microphysics scheme within the UM was found to hinder its ability to reproduce tenuous cloud
periods during the *Arctic Summer Cloud Ocean Study* (ASCOS); when clouds were modelled, the model produced too thin
cloud layers in a boundary layer (BL) that was often too well-mixed and too shallow (**Birch et al., 2012**). The prevalence of
too much low-level cloud caused surface energy balance, and hence surface temperature, biases. The new Cloud-Aerosol
Interactive Microphysics (CASIM) double-moment scheme in the UM has enabled improvements in its representation of Arctic
clouds; **Stevens et al. (2017)** noted it improved the surface net longwave radiation ($LW_{net}$) in both cloudy and cloud-free
conditions. Specifically, inclusion of aerosol processing within CASIM successfully led to cloud dissipation when modelling
the CCN-limited clouds observed during the ASCOS campaign, indicating that this explicit description of double-moment
microphysics (rather than a simplified cloud physics description) is key to modelling these clouds in the high Arctic.
Like the UM, the IFS also failed to capture episodic cloud-free periods observed during ASCOS, leading to similar surface
energy biases (**Sotiropoulou et al., 2016**). The updated IFS cloud scheme, used operationally since 2013, has improved its
ability to capture mixed-phase Arctic clouds in recent revisions; however, **Sotiropoulou et al. (2016)** reported that the IFS still
exhibits a persistent positive near-surface temperature bias, despite the improvement to its representation of these clouds. These
Arctic surface biases persist in version Cy45r1 of the model, as shown by **Tjernström et al. (2021)**. Given that reanalysis
products created using the ECMWF IFS (e.g., ERA5; **Hersbach et al. 2020**) are widely used, both to produce lateral boundary
conditions for process studies with numerical weather prediction (NWP) models and to analyse Arctic atmospheric structure,
we must understand the root of these biases and make recommendations for process improvements.
Here, we evaluate the performance of recent revisions of both the UM and IFS focusing on their ability to capture clouds and
the thermodynamic structure of the BL, highlighting common process relationships between the models which may explain
differences from observations. To achieve this, we compare these models with recent high Arctic observations made during the
Arctic Ocean 2018 (AO2018; **Vüllers et al., 2021**) expedition, where a suite of remote-sensing instrumentation was active
aboard the Swedish icebreaker *Oden* measuring summertime cloud and BL properties in the high Arctic. We use Cloudnet
(**Illingworth et al., 2007**) to compare observations with cloud properties simulated by the models, to test the respective
components in each model simulation with a focus on evaluating the relative contributions of the following on cloud structure:

1.   The choice and use of large-scale cloud schemes at high resolution

2.   The cloud microphysics scheme chosen to represent resolved clouds

3.   Representative CCN concentrations, and thus droplet number concentrations, as a function of altitude

4.   The global model analyses used to produce boundary conditions for high resolution nested configurations

By testing these hypotheses with two different atmospheric models, operating on different grid configurations, we assess
whether representative CCN concentrations are indeed the key model development still required to suitably capture Arctic
clouds, or whether other factors are restricting model performance in the high Arctic.





**2**      **Data and methods**
**2.1**      **Arctic Ocean 2018 expedition**
During the AO2018 expedition, *Oden* drifted with an ice floe near the North Pole from 14 August to 14 September 2018 (**Fig.**
**1**). Campaign details, instrumentation, and meteorological measurements from the AO2018 expedition are summarised in
**Vüllers et al. (2021)**. Here, we use a subset of the measurements for direct comparison with our model simulations. Key
acronyms, abbreviations, and parameters referenced in this study are listed in **Table 1**.
Radiosondes (Vaisala RS92) launched at 0000, 0600, 1200, 1800 UTC, provide in situ thermodynamic profiles with a 0.5°C
and 5% manufacturer specified uncertainty associated with temperature and humidity sensors, respectively. The radiosonde
data were distributed via the global telecommunications system and assimilated operationally at the Met Office and ECMWF.
Remote sensing measurements from a Metek MIRA-35 Doppler cloud radar, a Halo Photonics Streamline Doppler lidar, and
an RPG HATPRO microwave radiometer were processed through the Cloudnet algorithm (**Illingworth et al., 2007**) following
the data preparation steps of **Achtert et al. (2020)**. A Vaisala PWD22 present weather sensor (PWS) measured visibility,
precipitation type, precipitation intensity, and cumulative amount; near surface temperature and relative humidity (RH) were
obtained from an aspirated Rotronic HMP101 sensor. Broadband downwelling solar and infrared radiation were measured on
board the ship by Eppley PSP and PIR radiometers. 3-hourly albedo estimates from surface images were used to calculate
upwelling shortwave radiation (**Vuellers et al., 2021**).
**2.2**      **Cloudnet**
Cloudnet is used to directly compare between our measured and modelled cloud properties (**Illingworth et al., 2007**). Cloudnet
ingests Doppler cloud radar and lidar, ceilometer, microwave radiometer, and radiosonde data to derive cloud fractions and
cloud water contents on a chosen model grid. A comprehensive description of the algorithm is beyond the scope of this paper
and is provided by **Illingworth et al., 2007 (and references therein),** but essentially the algorithm first homogenizes
observational data to a common time resolution of 30 s and interpolates data to the radar height grid. Radar reflectivity ($Z_e$) and
lidar backscatter ($\beta$) profiles are used to determine cloud boundaries. Cloudnet takes advantage of the lidar's sensitivity to
small particles, such as cloud droplets and aerosol, and the radar's sensitivity to large particles, such as ice particles, rain, and
drizzle. Cloud phase is determined using $Z_e$, $\beta$, and thermodynamic information from the radiosondes. Cloud ice water content
($IWC$) is derived using $Z_e$ and temperature (**Illingworth et al., 2007**), while liquid water content ($LWC$) is derived by
partitioning $LWP$ measured by the radiometer to the identified liquid cloud layers from the lidar. Additionally, an
adiabatic $LWC$ is calculated from temperature and humidity profiles and the identified cloud top and base height from radar
and lidar measurements.
Cloudnet has already been utilised to study Arctic cloud properties using measurements made aboard *Oden* both during this
campaign (**Vüllers et al., 2021**) and during the *Arctic Clouds in Summer Experiment* in 2014 (**Achtert et al., 2020**). Potential
errors associated with the Cloudnet procedure are described in **Achtert et al. (2020)**. One particular limitation relevant to this
study is the minimum detection height of 156 m (lowest radar range gate). Low level clouds/fog below this height are hence
missed by Cloudnet (**Vüllers et al., 2021**) and not included in model comparisons. This limitation also results in problems with



the $LWC$ derived from radiometer measurements; therefore, we use the calculated $LWC$ under adiabatic assumption in this
study (for further details see **Appendix A**)
For comparisons with models, cloud fraction by volume ($C_V$), adiabatic $LWC$, and cloud $IWC$ from observations are averaged
to a reference model grid; here, we use the UM grid, but we could have equally chosen that of the IFS (**Fig. S1**). Cloud properties
are calculated using measurement profiles alongside model wind speed and grid-box size, where changes in cloud properties
over time are assumed to be driven primarily by advection and not microphysical changes (**Illingworth et al., 2007**). This
procedure is applied for $C_V$, $LWC$, and $IWC$, with $C_V$ defined as the fraction of pixels in a 2D slice which are categorised as
liquid, supercooled liquid, or ice (**Illingworth et al., 2007**).

### 2.3 Models

A summary of each model simulation is included in **Table 2** and detailed in the following sections. 36-hour forecasts were
performed with each model, initialised each day at 1200 UTC with the first 12 hours of spin up discarded, thereby producing
daily forecast products (0000 UTC – 0000 UTC) with hourly diagnostics for analysis. This is common practice for such
forecasts to ensure discrepancies due to spin up are avoided, while maintaining meteorology close to reality; however, as noted
by **Tjernström et al. (2021)**, model error growth is often a function of forecast time and thus the findings of this study may be
related to the time window chosen for each model forecast.
Column diagnostics from the grid cell closest to the position of *Oden* were extracted from the model domain, updated hourly
to account for the ship's drifting position. These variables (e.g., temperature, humidity, cloud fraction, condensate variables,
wind versus time) were then used for comparisons with alike variables constructed using Cloudnet (**Illingworth et al., 2007**)
with measured data (see **Sect. 2.1**).

### 2.3.1 Integrated Forecasting System (IFS; ECMWF_IFS)

Cycle 46r1 (Cy46r1) of the IFS (used operationally from Jun 2019 to Jun 2020) was used to create global meteorological
forecasts. The IFS uses a spectral formulation with a wave-number cut off corresponding to a horizontal grid size of
approximately 9 km (**Fig. 1b**). It has 137 levels in the vertical up to 80 km, the lowest at ≈10 m with 8 levels below ≈200 m
and 20 below 1 km. IFS forecasts were initialised from ECMWF operational analyses. Operational forecasts produced at the
time of the campaign (with Cy45r1) were recently evaluated on a 3-day lead time from a statistical viewpoint for this expedition
(**Tjernström et al., 2021**); in contrast, lead-time averaged verification was conducted in this study using a 1-1 comparison of
a concatenated timeseries of forecast values (T+13—T+36) with hourly observations.
Cloud properties are parameterised following **Forbes and Ahlgrimm (2014)**. This cloud scheme was implemented in Cy36r4
and has been previously evaluated for Arctic clouds by **Sotiropoulou et al. (2016)** using Cy40r1. Five independent prognostic
cloud variables are included (grid box fractional cloud cover, and specific water contents for liquid, rain, ice, and snow).
Heterogeneous primary ice formation is diagnosed following **Meyers et al. (1992)**, with a mixed-phase cut-off of –23 °C.
Liquid cloud formation occurs when the average relative humidity within a grid box exceeds a critical threshold, $RH_{crit}$,
representing sub-grid-scale variability of moisture. This threshold is 80% in the free troposphere, increasing towards the surface





in the boundary layer (**Tiedtke, 1993**). Once formed, cloud liquid mass is distributed across a fixed cloud droplet number
concentration, $N_d$, of 50 cm$^{-3}$ over the ocean (and 300 cm$^{-3}$ over land) to act as a threshold for autoconversion from liquid to
rain. For interactions with the radiation scheme, the IFS follows **Martin et al. (1994)** for estimating droplet number, using the
prognostic specific liquid water content and a prescribed CCN profile. CCN concentrations are calculated as a function of the
near-surface wind speed but decreases with altitude to represent the vertical distribution of aerosol within and above the BL.
Further details regarding the cloud scheme can be found in the ECMWF documentation (**IFS Documentation – Chapter 7:**
**Clouds and large-scale precipitation, https://www.ecmwf.int/node/19308**).
The IFS is coupled to a 0.25° resolution dynamic sea-ice model (Louvain-la-Neuve Sea Ice Model, LIM2) which provides sea
ice fractions to the IFS and the surface flux tiling scheme (**Buizza et al., 2017; Keeley and Mogensen, 2018**). The surface
energy balance over the sea ice fraction is, however, calculated separately from LIM2 using an albedo parameterisation
following **Ebert and Curry (1993)** with fixed monthly climatology values interpolated to the actual time, and a heat flux
through the ice calculated using a constant sea-ice thickness of 1.5 m.

### 2.3.2    Unified Model (UM)

The UM was operated as a high-resolution LAM with a 1.5 km × 1.5 km grid (grid is shown in **Fig. 1**). A rotated pole
configuration provided approximately equal spacing between grid points towards 90 °N. The LAM contained 500 × 500 grid
boxes, spanning from 83.25 °N to 90 °N centred on the 30 °E meridian. In the vertical, there were 70 vertical levels up to 40
km, with 24 levels within the lowest 2 km of the domain (**Grosvenor et al., 2017**). Lateral boundary conditions were generated
hourly from UM global model 36-hour forecasts at N768 resolution (corresponding to approximately 17 km at 90 °N with the
rotated pole) using the Global Atmosphere 6.1 configuration (**Walters et al., 2017**; **Table 2**). Three configurations of the UM
LAM were tested for the main body of this study, each using different combinations of cloud microphysics and large-scale
cloud schemes. Details on the pertinent microphysical processes represented in each simulation are listed in **Table 3**.

### 2.3.2.1   Regional Atmosphere model configurations (UM_RA2M and UM_RA2T)

Version 2 of the Regional Atmosphere model within the UM framework has two standard configurations: the mid-latitude
configuration (UM_RA2M) and the tropical configuration (UM_RA2T). Both are used operationally in their respective
geographical regions. The key difference between these configurations can be found in their turbulent mixing processes:
UM_RA2M employs weak turbulent mixing to encourage heterogeneity in model fields to facilitate the triggering of small
convective showers; however, while this weak mixing works well to reproduce conditions often experienced in the mid-
latitudes, it triggers convection too early in the tropics. Therefore, these two standard Regional Atmosphere configurations
were designed separately to account for these subtle differences in convection initiation on km-scales (**Bush et al., 2020).**
Neither configuration has been previously evaluated for use in the Arctic. Note that both UM_RA2M and UM_RA2T use the
default Regional Atmosphere surface albedo thresholds, giving a 50% albedo at an ice surface temperature of 0 °C and
increasing to 80% at –10 °C. **Gilbert et al. (2020)** tested both configurations for polar cloud modelling over the Antarctic
Peninsula, finding that the mid-latitude scheme performs better than the tropical configuration for capturing polar cloud liquid





water properties and associated radiative interactions (with the surface albedo modelled to within 2% of observed values),
whereas the tropical scheme enabled a too-efficient ice phase (and associated liquid depletion).
Both UM_RA2M and UM_RA2T include the **Wilson and Ballard (1999)** description of large-scale precipitation to simulate
resolved cloud microphysics. This microphysics scheme describes prognostic liquid and ice mixing ratios ($q_{liq}$ and $q_{ice}$,
respectively), with an assumed fixed $N_d$ profile calculated from an aerosol climatology and tapered to 50 cm$^{-3}$ towards the
surface (between 150 m and 50 m). A single ice species (encapsulating pristine crystals, aggregates, and snow particles) is
represented, with an assumed particle size distribution based on **Field et al. (2007)**.
UM_RA2M uses the **Smith (1990)** large-scale cloud scheme to parameterise sub-grid-scale fluctuations in humidity and cloud,
designed to ensure coarse grid GCMs do not have entirely cloudy grid boxes. It diagnoses cloud fraction and condensate
variables for input to the microphysics scheme, referencing a prescribed $RH_{crit}$ profile (based on a symmetric triangular
probability density function of sub-grid-scale variability in temperature and moisture) to permit condensation below 100 %
humidity (**Wilson et al., 2008**). Condensation cannot occur within a grid box until the grid box mean $RH$ exceeds a prescribed
altitude-dependent $RH_{crit}$ (described in **Table 3**).
In UM_RA2T, the prognostic cloud fraction and prognostic condensate (PC2; **Wilson et al., 2008**) large-scale cloud scheme
is used, designed to address the over-sensitive diagnostic links between cloud fraction and cloud condensate in **Smith (1990)**.
Total, liquid, and ice cloud fractions are included as prognostic variables in PC2, allowing cloud fractions and condensate to
vary through other interactions (such as BL processes and cloud microphysics) and not simply diagnosed from temperature and
humidity as in **Smith (1990)** (**Wilson et al., 2008**). PC2 prognostic variables are advected by the wind and continually updated
following incremented sources and sinks in the model, with the additional inclusion of sub-grid-scale turbulent production of
liquid in mixed-phase cloud from an analytical model of sub-grid-scale moisture variability (**Furtado et al., 2016**). Differences
between the methods of representing cloud fraction in the PC2 and Smith schemes are detailed in the **Supporting Information**.
**2.3.2.2    Regional Atmosphere with Cloud-Aerosol Interactive Microphysics scheme (UM_CASIM-100)**
UM_CASIM-100 uses the CASIM scheme (detailed by **Hill et al., 2015**) coupled with the **Smith (1990)** large-scale cloud
scheme (as in **Grosvenor et al., 2017**). **Stevens et al. (2017)** previously tested the CASIM scheme within the UM nesting suite
in an Arctic cloud case study, showing that it performed well in capturing cloud dissipation; however, the authors did not
include sub-grid-scale contributions from **Smith (1990)** in that study.
CASIM utilises prescribed lognormal aerosol distributions to provide a double-moment representation of cloud particle
processes and is the only double-moment setup included in this study. Particle size distributions of five hydrometeors (liquid
droplets, ice, snow, graupel, and rain) are each described by a gamma distribution, with prognostic mass mixing ratios and
number concentrations. Ice number concentrations are diagnosed via a temperature-number concentration parameterisation
(**Cooper, 1986**), but require liquid to be present before ice can form; a relationship thought to be important in Arctic mixed-
phase clouds (e.g., **de Boer et al., 2011; Young et al., 2017**). Droplet activation follows **Abdul-Razzak and Ghan (2000)**,
referencing a fixed soluble accumulation mode aerosol number concentration profile of 100 cm$^{-3}$. This profile was



approximated based on aerosol concentration profiles previously measured during summertime in the central Arctic
(**Kupiszewski et al., 2013**).
CASIM offers user flexibility regarding aerosol processing, as described by **Miltenberger et al., (2018)**. Here we do not impose
wet-scavenging processes, likely important for capturing cloud-free conditions, for consistency with the simpler single-moment
liquid microphysics schemes used in the other simulations; however, use of this option will be explored in future work.
For our CASIM simulation, we adapt the warm ice temperature albedo of the LAM to 72% (at 0 °C), with 80% albedo achieved
at –2 °C, to match the parameterisation limits currently used in the *Joint UK Land Environment Simulator* surface scheme of
the Global Atmosphere 6.0 global model (under the assumption that snow is present on the sea ice surface). For the drift period,
we know that snow was indeed present on the surface from first-hand knowledge and surface imagery; therefore, we use this
simulation to test the effect of such an increased albedo at warmer surface ice temperatures on the modelled surface energy
balance. An example simulation utilising the CASIM scheme with the default Regional Atmosphere albedo settings used in
UM_RA2M and UM_RA2T, to demonstrate the radiative impact of CASIM alone, is described in the **Supporting Information**
**(Sect. S2).**
**2.4        Comparison methodology and compared parameters**
$C_V$, $q_{liq}$, and $q_{ice}$ from each model simulation were ingested by Cloudnet to calculate $LWC$ and $IWC$. Within these calculations,
Cloudnet filters model data for values outside the range observable by the instrumentation used; for example, $q_{ice}$ data are
filtered for values which would be beyond the observable range of the radar.
We use an additional metric alongside $C_V$ based on total condensate for comparisons between our measured and modelled
clouds; a total water content ($TWC$) mask where the grid-box is considered cloudy; this mask is set to 1, when $TWC \geq 1 \times 10^{-6}$
$^6$ kg m$^{-3}$ below 1 km, and $TWC \geq 1 \times 10^{-7}$ kg m$^{-3}$ above 4 km, with vertical interpolation in between (following **Tjernström et**
**al., 2021; Fig. S3**). While this mask will not capture fractional cloud at cloud boundaries, averages of this mask are directly
comparable between the observations and models. It acts as a comparison metric based solely on cloud water contents, which
are prognostic in every simulation, and does not depend on a specific definition of e.g., cloud fraction.
In addition to a full overview of model performance over the drift, we further split our data into sub periods to aid our
interpretation of the comparisons between the measurements and models. The sea ice melt/freeze transition was captured by
the measurements; **Vüllers et al. (2021)** identified the sea ice freeze onset date as 28 Aug and defined sub-periods throughout
the drift based on consistent meteorology (see **Fig. 2g**). We concentrate on the sea ice melt and freeze periods separately and
on shorter episodes within these periods; one during the sea ice melt (14—18 Aug) and one during the freeze (4—8 Sept).
**3        Results**
**3.1        Surface radiation**
**Figure 2** shows measured and modelled time series of net surface shortwave ($SW_{net}$) radiation, $LW_{net}$, and the combined
surface net radiation ($R_{net}$) during the AO2018 drift period. All radiative quantities are defined as positive downwards.


All models overestimate $SW_{net}$ (**Fig. 2a**) with respect to measurements, with ECMWF_IFS and UM_CASIM-100 in better
agreement with observations than UM_RA2M and UM_RA2T. From **Fig 2d**, all simulations fail to capture strong longwave
net emission likely related to cloud-free episodes (e.g., 20—21 Aug) and sporadically predict such cloud-free conditions (and
net longwave emission) when clouds were observed (e.g., 2 Sep).
Considering the melt and freeze periods separately, the measured $R_{net}$ is often negative after 28 Aug (**Fig. 2g**) driven by $LW_{net}$,
while $SW_{net}$ decreases with the declining solar elevation angle (**Fig. 2a**). In contrast, the models' net radiation is not typically
negative until after 8—9 Sep, excluding a short negative period at 2 Sep driven by the lack of modelled cloud (as suggested by
strong net longwave emission; **Fig. 2d**). This delay would likely affect the freeze onset if the models were fully coupled to a
sea ice model; as such, this feedback may be active within the (simple) coupled atmosphere-sea ice system of the IFS.
Probability Density Functions (PDFs) of these data, split between melt and freeze periods (**Fig. 2b—c, e—f, h—i**), reveal some
clear distinctions in model capability. $SW_{net}$ PDFs vary substantially between the models during the melt period (**Fig. 2b**); no
simulation captures the observation distribution well. Observed $SW_{net}$ from the ship has a median of +18.2 W m$^{-2}$, with each
simulation producing medians at greater values (UM_CASIM-100 = +18.7 W m$^{-2}$; ECMWF_IFS = +21.6 W m$^{-2}$; UM_RA2M
= +40.5 W m$^{-2}$; and UM_RA2T = +41.9 W m$^{-2}$). While the medians for UM_CASIM-100 and ECMWF_IFS are in good
agreement with observations, both exhibit a too-narrow distribution. These too-narrow distributions – which also all lack a very
high positive tail – suggest that the modelled cloud cover is too consistent, likely related to the lack of cloud-free episodes
indicated by the $LW_{net}$ data (**Fig. 2d**). Median $SW_{net}$ of both the UM_RA2T and UM_RA2M PDFs is much too high, with
non-negligible occurrences > +50 W m$^{-2}$. The improvement of UM_CASIM-100 over UM_RA2T and UM_RA2M indicates
that both the surface albedo used by default in the Regional Atmosphere configurations is too low and the updated cloud physics
description of CASIM improves the modelled cloud-radiation interactions. A trial simulation utilising the cloud physics setup
of UM_CASIM-100 alongside the default Regional Atmosphere surface albedo parameterisation inputs (as used in UM_RA2M
and UM_RA2T) shows that the double-moment cloud physics representation alone does improve radiative properties with
comparison to the standard configurations (see **Supporting Information**); however, the combination of improved cloud-
radiation interactions and an updated surface albedo (as shown here in UM_CASIM-100) provides the best agreement between
the UM and our observations.
During the freeze period, measurement estimates of $SW_{net}$ peak at +7.9 W m$^{-2}$, while ECMWF_IFS, UM_CASIM-100,
UM_RA2M, and UM_RA2T have maxima at +10.0, +10.4, +25.0, and +26.6 W m$^{-2}$ respectively (**Fig. 2c**). The peak modelled
$SW_{net}$ remains too high in all simulations but, in contrast to the melt period, all PDFs are now too broad. ECMWF_IFS and
UM_CASIM-100 perform best with comparison to observations (both with a positive bias of less than +3 W m$^{-2}$ at their peaks).
However, both UM_RA2M and UM_RA2T have a broad bimodal structure, with the secondary peak in better agreement with
the observations than their maxima. Both UM_RA2T and UM_RA2M are largely in better agreement with observations during
the freeze period than during the melt; this improved agreement is likely due to either a better representation of incoming
shortwave radiation or the surface albedo; the surface temperatures decreases through the transition to sea ice freezing
conditions, and **Fig. S4** indeed shows that the albedo modelled during the freeze for UM_RA2M and UM_RA2T is in better
agreement with observational estimates than that modelled during the melt period.





During the melt period, $LW_{net}$ aligns well between the measurements and models; however, all simulations produce a narrower
PDF than the observations and largely miss the tail $<$ —20 W m$^{-2}$ (**Fig. 2e**) resulting from observed cloud-free episodes at 15—
16 Aug, 20 Aug, 22 Aug, and 26 Aug (**Fig. 2d**). Despite this, each simulation performs well in replicating the median of the
PDF, with a maximum model-observation difference of —1.9 W m$^{-2}$ (UM_RA2T). As with $SW_{net}$, model-observation
agreement generally improves during the freeze period, with UM_RA2M, UM_RA2T, and ECMWF_IFS producing PDFs
closely matching the observations, with median values at —9.4, —11.5, and —6.8 W m$^{-2}$ respectively compared with an
observation peak of —6.5 W m$^{-2}$ from the ship estimates. Each of these cases also reproduces the negative distribution tail
missed by all simulations during the melt (**Fig. 2f**). UM_CASIM-100 displays a narrower distribution with fewer negative
values, yet still performs equally well in reproducing the median of the $LW_{net}$ PDF (with a bias of —5.5 W m$^{-2}$). With the
exception of the too-narrow UM_CASIM-100 PDF, this improved agreement in $LW_{net}$ indicates that cloud cover is indeed
captured better by the models during the freeze, and remaining discrepancies in the $SW_{net}$ comparisons may indeed be related
more so to cloud microphysical structure or surface properties.
To investigate this relationship in more detail, we split our radiation data into the periods of consistent meteorology indicated
on **Fig. 2g.** In agreement with **Fig. 2**, model $SW_{net}$ and downwelling shortwave radiation ($SW_\downarrow$) biases are at their greatest
during period 3 (**Table 4**). All simulations similarly exhibit their greatest $LW_{net}$ biases during period 3 (**Table 4**); less cloud
cover was observed during this period with relation to other periods during the drift (**Vüllers et al., 2021**). Both UM_RA2T
and UM_RA2M perform best in terms of $SW_\downarrow$ during period 3; however, both perform particularly poorly in $SW_{net}$, indicating
that their surface albedo is not representative of observations. During period 4, UM_CASIM-100 performs very well in terms
of these $SW_\downarrow$ biases, suggesting the modelled cloud structure was in good agreement with observations. However, ECMWF_IFS
exhibits the smallest $SW_\downarrow$ biases of the four simulations during periods 5 and 6; periods when the sea ice was beginning to
freeze.
Each of these simulations highlight that small $SW_\downarrow$ biases do not necessarily produce similarly small $SW_{net}$ biases, as both the
modelled cloud properties and surface albedo need to be representative to remedy the $SW_{net}$ discrepancies. In UM_RA2T and
UM_RA2M, the surface albedo is poorly captured, as indicated by the consistently high $SW_{net}$ biases; however, ECMWF_IFS
and UM_CASIM-100 perform better in terms of surface albedo, with UM_CASIM-100 performing the best with the smallest
$SW_{net}$ biases across the four sub-periods considered. Further discussion of the surface albedo comparison is included in the
**Supporting Information**.
$LW_{net}$ biases do not exceed +5.5 W m$^{-2}$ over periods 4—6; however, biases are greater (up to +16.3 W m$^{-2}$ during period 3;
**Table 4**) due to the models' inability to reproduce cloud-free conditions. This relationship with cloud cover influences the
surface downwelling longwave ($LW_\downarrow$) biases: with the exception of the standard UM configurations during period 5, all $LW_\downarrow$
biases are positive (**Table 4**).
Combining these radiative components, we find that $R_{net}$ is overestimated by all simulations during the melt (with
UM_CASIM-100 and ECMWF_IFS performing better than UM_RA2M and UM_RA2T; **Fig. 2h**), largely driven by too much
surface $SW_{net}$ when cloud is present in reality, thus indicating that the model surface albedo is too low and thus does not reflect
enough $SW_\downarrow$. On the other hand, there are also non-negligible occurrences of too much modelled cloud when the conditions





should be cloud-free, driving strong $LW_{net}$ biases at these times. While agreement with observations largely improves during
the freeze period, these discrepancies still exist in the $SW_{net}$ data. While the $SW_{net}$ biases may be strongly influenced by errors
in the surface albedo, and thus beyond the scope of this study, the role of cloud structure in $SW_{\downarrow}$ biases and the $LW_{net}$ emission
episodes missed by each simulation are driven by the description of cloud: in the following sections, we investigate the cloud
macro- and microphysical structure and surface properties to explain these radiative differences.
**3.2**      **Cloud properties**
To evaluate model performance, we use two metrics for cloud occurrence: the model diagnosed cloud fraction, $C_V$, and the
cloud occurrence inferred from cloud water contents, the $TWC$ cloud mask. **Figure 3** shows $TWC$ and cloud fraction, $C_V$,
calculated from observations using Cloudnet and output by models. $TWC$ comparisons indicate that each simulation captures
the observed cloud aloft, except for UM_CASIM-100 between 4 and 10 Sep. Below 3 km, observed $TWC$ is generally lower
in magnitude than the model simulations.
In contrast, all simulations except UM_RA2T fail to reproduce the observed $C_V$ aloft. Low-altitude (below 2 km) cloud cover
appears to be captured comparatively better across all simulations. Cloud height simulated by ECMWF_IFS is in reasonable
agreement with the observations; however, there are notable periods where the persistence of clouds aloft is not reproduced.
For example, the altitude and timing of onset of the (likely precipitating) high clouds at 3—4 Sep is initially captured, but the
clouds are not sustained. Cloud layers aloft appear more tenuous also in UM_RA2M and UM_CASIM-100 than in the
observations: there are few cases of cloud fractions > 0.5 at altitudes above 3 km.
**Figure 4a** shows mean profiles of $C_V$ over the drift period. Only periods where we have measurement data are included in these
profiles for fair comparison. Note that cloud fraction below 0.15 km is not evaluated against observations here due to low-
altitude measurement limit of the cloud radar. Supporting qualitative interpretation of **Fig. 3**, model-observation agreement of
$C_V$ is best at low altitude (below 1 km); however, all simulations produce too much very low (between 0.15 and 0.5 km) cloud.
Modelled near-surface $C_V$ (between 0.15 and 0.5 km) is up to 16% too high (UM_RA2T). However, we can speculate that the
frequent fog episodes reported during the ice drift **(Vüllers et al., 2021)** may be somewhat captured by the models, as indicated
by mean values of $C_V$ below 0.15 km of 82%, 72%, 53%, and 39% respectively for UM_RA2T, UM_RA2M, UM_CASIM-
100, and ECMWF_IFS. All simulations except UM_RA2T perform poorly aloft: ECMWF_IFS, UM_RA2M, and
UM_CASIM-100 strongly underestimate $C_V$ between 1 km and 8 km, with UM_CASIM-100 and UM_RA2M reproducing less
than 20% of the observed $C_V$ at 4.5 km. Only the UM_RA2T $C_V$ profile agrees well at altitude, with particularly good agreement
between 0.5 and 2 km. In fact, $C_V$ between 2 km and 5.5 km agrees best with observations out of the four simulations considered.
**Figure 3** highlights that the observations, UM_RA2T, and (to an extent) ECMWF_IFS have a $C_V$ field scaling largely as either
0 or 1, whereas UM_RA2M and UM_CASIM-100 are more likely to have a fractional cloud cover aloft, thus producing a poor
comparison with our observations **(Fig. 4a)**. Despite this, qualitative model-observation comparisons of $TWC$ indicate that the
models are performing well. Further discussion of these differences is included in the **Supporting Information**. In summary,
the Cloudnet calculation of $C_V$ from observations is not directly equivalent to our model cloud fractions and such comparisons,
in isolation, should be approached with caution in the Arctic. To bypass this issue, we also use a cloud mask built from $TWC$





368 data to aid interpretation of our results. The observed $TWC$ cloud mask (**Fig. 4b**) differs from the mean $C_V$ profile, with a subtle

369 bimodal structure peaking at approximately 0.5 and 4.5 km (with a minimum around 2 km).

370 All simulations overestimate cloud occurrence below 2.5 km (**Fig. 4b**), in contrast to the underestimation between 1 and 2.5

371 km shown in the $C_V$ data (**Fig. 4a**). Mean observed cloud occurrence only reaches 75% between 0.15 km (lowest radar range

372 gate) and 0.5 km, while UM_RA2M and UM_CASIM-100 have more than 98% cloud occurrence at 0.2 km. UM_RA2T

373 performs slightly better, peaking to only 92% at 0.2 km; however, the improvement is not as significant between UM_RA2T

374 and UM_RA2M/UM_CASIM-100 as is suggested by the mean $C_V$ profiles (**Fig. 4a**). ECMWF_IFS peaks at a slightly higher

375 altitude, overestimating cloud occurrence by 33% at approximately 0.5 km (**Fig. 4b**).

376 Above 2 km, ECMWF_IFS, UM_RA2M, and UM_RA2T perform similarly; the greatest difference aloft occurs at 4.5 km,

377 where there is a minor peak in the mean observed cloud occurrence (up to 41%; **Fig. 4b**). ECMWF_IFS produces only 28%

378 cloud cover at this altitude. UM_CASIM-100 cloud occurrence monotonically decreases with altitude above 3.5 km, producing

379 only 20% cloud cover at 4.5 km, in agreement with the qualitative findings of **Fig. 3**. Therefore, with the exception of

380 UM_CASIM-100, the $TWC$ cloud masks indicate that modelled cloud occurrence aloft is, in fact, in reasonable agreement with

381 observations, in contrast to the trends indicated by the $C_V$ data (**Fig. 4a**). These data suggest that the $C_V$ comparisons may be

382 misleading if used in isolation, likely due to the different methods for representing cloud fractions and associated sub-grid-

383 scale variability in models (see **Supporting Information**).

384 Averaged in-cloud water content profiles are shown in **Fig. 4c—d**. Adiabatic $LWC$ calculated from observations with Cloudnet

385 is shown in **Fig. 4c**. This adiabatic assumption was used in place of the HATPRO $LWP$ due to the data quality issues introduced

386 to the latter because of the frequent occurrence of fog at altitudes below the lowest radar range gate (0.15 km; discussed further

387 in **Appendix A**).

388 The adiabatic $LWC$ peaks between 0.5 and 1 km then decreases steadily with altitude between 1 and 3 km. All simulations

389 overestimate in-cloud $LWC$ between 1 and 3 km; however, below 1 km, each simulation (except UM_RA2T) performs

390 reasonably well. At 0.5 km, UM_RA2T underestimates by 47 %, while UM_CASIM-100 overestimates by just 10 % and

391 UM_RA2M and ECMWF_IFS are in reasonable agreement with observations. UM_RA2T and UM_RA2M have bimodal

392 distributions, with peaks below 0.5 km and around 2 km, perhaps linked to their common use of the **Wilson and Ballard (1999)**

393 microphysics scheme. The increase in $LWC$ towards the surface in UM_RA2M is suggestive of fog, and UM_RA2M is the

394 only simulation to display this vertical structure. The mean $LWC$ calculated for ECMWF_IFS is consistent with altitude

395 between 0.5 and 2 km; however, there is more variability at 2 km than at lower altitudes, indicating that this may be a more

396 dominant liquid cloud layer at some time periods. Only UM_CASIM-100 displays a similar shape to the observations, yet its

397 $LWC$ is often greater than the observed $LWC$ at all altitudes above 1 km.

398 All simulations agree with the Cloudnet-calculated $IWC$ above 4 km (**Fig. 4d**); in fact, UM_RA2M performs particularly well

399 across the entire vertical profile. ECMWF_IFS and UM_CASIM-100 also agree well for most of the profile apart from slight

400 overestimations below 1.5 km (though still within one standard deviation of the observed mean). UM_RA2T overestimates

401 below 4 km, producing almost seven times the observed $IWC$ (0.019 g m$^{-3}$ versus 0.003 g m$^{-3}$) at 0.5 km. Shaded standard

402 deviations also indicate that UM_RA2T is also more variable than both the three other simulations and the measurements,





consistent with previous studies showing its ice phase is more active than UM_RA2M in polar mixed-phase clouds (**Gilbert**
**et al., 2020).**
Column-integrated metrics and surface measurements provide an additional perspective for evaluating model performance with
regards to clouds. Measured $LWP$ and precipitation fluxes are shown alongside corresponding model diagnostics in **Fig. 5**.
Cloudnet-filtered $LWP$ is included in **Fig. 5a,b** for comparison; these data are HATPRO measurements filtered by Cloudnet
for bad points (e.g., strong precipitation events). ECMWF_IFS, UM_RA2M, and UM_CASIM-100 produce $LWPs$ in
reasonable agreement with measurements throughout the full drift period, with the PDFs of **Fig. 5b** indicating that these $LWPs$
are overestimated slightly with respect to the measurements/Cloudnet data. UM_RA2M overestimates in some periods, for
example the $LWP$ peak during the storm of 12 Sep is 230 g m$^{-2}$ more than measured (**Fig. 5a**). In contrast, UM_RA2T
underestimates the $LWP$ overall, with few occurrences of $> 200$ g m$^{-3}$ (**Fig. 5b**). This underestimation of $LWC$ (**Fig. 4c**) and
$LWP$ (**Fig. 5a, b**) by UM_RA2T aligns with its overestimation of $IWC$ below 4 km; with too much ice in mixed-phase cloud,
liquid is depleted too efficiently via the Wegener-Bergeron-Findeisen mechanism.
Each simulation broadly captures the notable precipitation events measured (**Fig. 5c—d**). UM_CASIM-100 and UM_RA2T
reproduce the measured total precipitation flux well and capture the short episodes of increased precipitation at 22 Aug, 3 Sep,
and 12 Sep. ECMWF_IFS and UM_RA2M also capture some precipitation events; however, the magnitude of these events is
best reproduced by UM_CASIM-100. No simulation reproduces the precipitation intensity measured at 8 Sep. While the key
precipitation events are largely captured by the models, with each model producing precipitation as predominantly snow rather
than rain, the precipitation rates simulated are low and likely contribute to the lack of cloud-free periods as indicated by the
$LW_{net}$ comparisons shown previously (**Fig. 2d, e, f**).
These results therefore indicate that the modelled microphysical structure is positively biased in terms of cloud liquid with
respect to observations (**Figs. 4c, d—5**). There is a consistent model-observation bias, with all simulations producing too much
cloud (**Fig. 4a, b**) below 2.5 km. In ECMWF_IFS, UM_RA2M, and UM_CASIM-100, this cloud contains too much liquid (as
indicated by positive biases in $LWC$ and $LWP$). Only UM_RA2T underestimates the cloud liquid properties due to its active
ice phase. **Figure 6** links the radiation, $LWP$, and $C_V$ biases of our four model simulations with respect to observations. $C_V$
biases are calculated as the model-observation bias below 3 km, where model data below the height of the lowest radar range
gate (0.15 km) are excluded from the bias calculation. Here, $C_V$ is used in place of the $TWC$ cloud mask as the latter is calculated
from in-cloud $LWC$ and is therefore not strictly independent of $LWP$. These linear regressions demonstrate that the positive
downwelling radiative biases are indeed tied to too much cloud cover within the models, and too much liquid within the
modelled clouds. The correlations are weaker for $R_{net}$, likely due to the additional influence of other factors (e.g., surface
albedo) on the net radiative properties.

### 3.2.1  Influence of CCN concentration

Each simulation overestimates cloud occurrence below 2.5 km and struggles to maintain cloud-free conditions, problems
previously identified for earlier versions of these models. Both **Sotiropoulou et al. (2016)** and **Birch et al. (2012)** commented
on the need for variable, representative cloud nuclei concentrations in the IFS and the UM to enable cloud-free periods to be
captured. A fixed accumulation-mode aerosol number and mass concentration profile was used in UM_CASIM-100; however,





such consistency with altitude is unlikely to occur in reality. While the concentration chosen was based on previous
measurements in the Arctic (**Kupiszewski et al., 2013**), aerosol number concentrations are typically very low and
heterogeneous within the BL during the Arctic summer (**Mauritsen et al., 2011; Tjernström et al., 2014**) yet long-range
transport provides comparatively greater, more homogeneous concentrations aloft.
An additional simulation with the CASIM scheme was tested, using a more representative CCN vertical profile guided by
output from the UK Chemistry and Aerosol (UKCA; **Morgenstern et al., 2009**; **O'Connor et al., 2014**) global model. Details
on the UKCA model configuration used to obtain these aerosol data are included in the **Supporting Information**. Using
representative aerosol profiles as input to the CASIM scheme (with lower CCN concentrations within the lower troposphere
and greater concentrations within the free troposphere; denoted UM_CASIM-AeroProf) can affect the $SW_\downarrow$ as expected via the
associated influence on $N_d$ and $q_{liq}$ (**Fig. 7**). Low altitude (below 1 km) clouds have a significantly lower $N_d$ , < 25 cm⁻³, in
UM_CASIM-AeroProf than in UM_CASIM-100. This low $N_d$ is expected from periodic episodes of low CCN in the Arctic
BL (**Mauritsen et al., 2011**); cloud residual concentrations of up to 10 cm⁻³ were measured on board *Oden* during the AO2018
expedition (**Baccarnini et al., 2021**). However, despite the differences in $N_d$ between these two CASIM simulations, $q_{liq}$ does
not differ much as the simulated clouds are not heavily precipitating (and thus cloud lifetime is largely unaffected).). This
similarity is also displayed in the diagnosed cloud fractions, related to the comparatively unaffected $q_{liq}$. Despite the
consistency in cloud fractions and $q_{liq}$, the cloud albedo is subtly lowered (as fewer CCN are available) in UM_CASIM-
AeroProf, as shown by the $SW_\downarrow$ comparisons in **Fig. 7a—b**.

### 3.3    Thermodynamic structure

Differences between modelled and observed cloud properties are likely related to the thermodynamic structure of the
atmosphere and how well this is modelled. **Figure 8** shows temperature ($T$) and water vapour specific humidity ($q$) from
radiosondes and anomalies of each simulation with respect to these measurements. The altitude of the main inversion base
identified from the radiosondes is shown in black.
Each simulation is typically too cold with respect to observations at altitudes just above the main inversion (*left column*; **Fig.**
**8**): this anomaly is a consistent feature throughout the drift period and across models, however it is most prominent at the
beginning of the drift. These trends indicate that the altitude of the modelled temperature inversion capping the BL is too high,
likely driven by too much BL mixing and the associated too-deep cloud layers modelled in each simulation **(Fig. 4b).** Below
the observed inversion, the simulations are typically warmer than measured; for example, at 18 Aug all UM simulations have
a particularly strong bias (> 3 K) below the observed main inversion, with ECMWF_IFS exhibiting a similar, but smaller, bias.
Above approximately 3 km the $T$ biases are typically smaller in magnitude and variable in sign. All UM simulations display
similar differences with respect to the radiosonde measurements; for example, each UM simulation exhibits a strong $T$ bias up
to 4.4 K at 6.5 km during 9 Sep.
$q$ biases are typically small throughout much of the atmospheric column (*right column*; **Fig. 8**), with some instances of larger
biases. These stronger biases are not confined to the lowest 3 km as with the temperature data. Radiosonde humidity data up to
22 Aug are variable aloft, and this noise affects the biases calculated over this period. However, a strong moisture bias of >





0.90 g m$^{-3}$ is evident between 2 km and 4 km over 20—22 Aug in all UM simulations. Similarly, the dry bias (of up to 1.86 g
m$^{-3}$) across the UM simulations from 2—4 Sep is notable and is also present, to a lesser extent, in ECMWF_IFS (up to 0.82 g
m$^{-3}$).
When these data are simplified into median profiles (**Fig. 9**), the characteristic biases exhibited by the models become clearer.
**Figure 9(a, c)** shows that the $T$ biases are small above 4 km, with all UM simulations exhibiting a slight warm bias and
ECMWF_IFS exhibiting a slight cold bias. Similarly, moisture biases are negligible above 4 km in all simulations (**Fig. 9b, d**).
However, below 4 km strong biases emerge.
From the surface up to 0.5 km, there is a decreasing positive $T$ bias in all simulations. However, the positive surface $T$ bias is
reduced during the freeze period for UM_RA2M and UM_RA2T (from +0.28/+0.31 K to +0.20/+0.14 K, respectively) while
it intensifies from +0.52 K (+0.46 K) to +0.90 K (+0.56 K) for ECMWF_IFS (UM_CASIM-100) (**Fig. 9c**).
During the melt period, all simulations underestimate the temperature between 1 and 3 km, yet there is a clear bimodal structure
evident in each profile with secondary negative peaks at lower altitudes (**Fig. 9a**). ECMWF_IFS remains too cold across a
deeper layer than the UM simulations, between 0.4 and 3 km. Both the IFS and the UM exhibit strong (up to –1.54 K) biases
at 1.75 km. The negative $T$ bias layers at lower altitudes differ in height between the models, with ECMWF_IFS reaching –
0.94 K at 0.85 km while the UM simulations exhibit negligible positive biases at this height. The secondary peak in the UM
simulations is in fact lower in altitude, at 0.4—0.5 km**.** $T$ biases are smaller than during the melt period, reaching up to –1.06
K (ECMWF_IFS) between 0.65 km and 1 km, and the negative bias peak at 2 km seen previously is no longer present (**Fig.**
**9c**).
Similarly, each simulation exhibits a positive $q$ bias towards the surface. These biases change little between the melt and freeze
periods (**Fig. 9b, d**); ECMWF_IFS produces the greatest bias in both periods (+0.31 g m$^{-3}$ during both the melt and freeze),
while UM_RA2T produce the lowest (+0.24 g m$^{-3}$ and +0.10 g m$^{-3}$ during the melt and freeze, respectively). ECMWF_IFS is
too dry, as well as too cold, between 0.5 and 4 km, while the UM simulations are typically too moist (though variable; **Fig.**
**9b**).
There is less variability in the $q$ biases during the freeze period. The UM simulations in particular exhibit only small $q$ biases
above 0.5 km (**Fig. 9d**). ECMWF_IFS performs well above 2 km; however, similar to trends identified during the melt, it is
again too dry between 0.5 and 2km.
**Figure 10** similarly shows the median $T$ and $q$ biases modelled by UM_CASIM-100 and UM_CASIM-AeroProf over the
whole AO2018 drift period. Even though the clouds are likely more representative of the high Arctic environment in
UM_CASIM-AeroProf than UM_CASIM-100, the thermodynamic biases are largely unchanged from the approximated
aerosol input of UM_CASIM-100. We speculate that these biases would perhaps differ more so if the modelled clouds were
precipitating strongly in either simulation, thus affecting $q_{liq}$ and cloud lifetime. However, considering each of the UM LAM
configurations shown here, there is little variability in their thermodynamic biases despite the differences in their representation
of aerosol inputs, cloud microphysics, and large-scale cloud scheme. Interestingly, these biases are shared by the UM global
model (UM_GLM, shown in grey; **Fig. 9**) used to generate lateral boundary conditions for each LAM. UM_GLM exhibits





similar biases as its high-resolution LAM counterparts, suggesting that these thermodynamic biases are sourced from the
driving model itself.

### 3.3.1    Influence of the UM driving model

To investigate how much the large-scale forcing is influencing the UM biases, an additional test was performed over a subset
of the drift (31 Aug to 5 Sep) using ERA-Interim to initialise the UM global model (labelled UM_RA2M-ERAI-GLM; **Fig**
**11**). This test was designed to evaluate whether the initial conditions of the global driving model, and therefore the associated
data assimilation (DA) systems used to derive the operational analyses used for initialisation, are largely responsible for the
LAM thermodynamic biases we have found in this study. For this test, we used the UM_RA2M configuration for the LAM,
and all global model physics options remained the same as in previous simulations (as described in **Table 2**); the only difference
was in the initial conditions of the global model.
We find that UM_RA2M-ERAI-GLM exhibits $T$ and $q$ biases following ECMWF_IFS between the surface and 3 km, inheriting
the ECMWF_IFS near-surface temperature bias discussed previously (**Fig. 11a**). Over this short time period, the UM
simulations do not have this bias. Above 3 km, UM_RA2M-ERAI-GLM follows UM_RA2M and UM_GLM, exhibiting a
slight warm bias (0.45 K at 5.5 km) in contrast to the cold bias of ECMWF_IFS (–0.65 K at the same altitude).
These results confirm that the UM LAM biases within the lower atmosphere shown in **Figs. 8** and **9** are driven by biases in the
large-scale forcing from the global model, which may be a result of the model physics itself or the DA used to produce the
operational analyses. Given that the Arctic lacks good observational data coverage, DA systems still rely heavily on their model
components when creating the analysis products used for model initialisation; therefore, improved observational data coverage
may improve these biases in the DA systems and thus global model initial conditions. In the meantime, a different LAM
configuration, with a larger nested domain with lateral boundaries further from the science region of interest may break the
relationship between global model and LAM biases shown here.

### 3.4    Links between cloud properties and thermodynamic biases

To better understand how the model thermodynamic biases relate to cloud properties in each simulation, we split our drift
period further into four subsections – periods 3 to 6, as illustrated in **Figs. 2 and 8** – to study periods of consistent meteorology.
Mean equivalent potential temperature ($\theta_e$) and $q$ profiles measured by radiosondes during these periods are shown in **Fig. 12**.
Of the four periods considered, period 3 had cloud-free conditions most often. Periods 5 and 6 were similar; both were cloudy
and influenced synoptically by three different low-pressure systems over their duration.
Cloud properties and thermodynamic biases during periods 3 and 6 are shown in **Fig. 13** (with similar analysis for periods 4
and 5 included as **Fig. S7**). As mentioned previously, mean observed cloud occurrence was lower for period 3 than in any other
period during the drift. All simulations overestimate the $TWC$ cloud mask below 2 km, with each UM case producing a bimodal
mean profile peaking below 0.5 km and at 1.8 km (**Fig. 13a**). Such bimodality is less clear with ECMWF_IFS; it exhibits a
lower layer with cloud top at 1 km and a more prominent than a secondary layer at 1.6 km, although the separation of these
layers is not as distinct as in the UM cases. The secondary layer at 1.6 km has a greater $LWC$ than the lower layer, with a peak





of 0.14 g m$^{-3}$ (**Fig. 13b**). The bimodal cloud structure is also liquid dominated in the UM simulations, where both peaks reach
around 0.1 g m$^{-3}$ (and even exceed this magnitude in the 1.8 km layer), across all three configurations.
Considering the corresponding median $T$ biases (**Fig. 13d**), there are clear correlations between negative biases and modelled
cloud height, suggesting that cloud top $LW$ cooling is a contributing source of these biases. The lower layer (0.75 km) bias in
ECMWF_IFS is particularly striking, reaching –4.45 K, and corresponds with the top of a large fraction of liquid-dominated
cloud (**Fig. 13a, b**). The mean $LWC$ modelled at this altitude is over three times greater than was observed, with cloud frequency
overestimated by 73%. $q$ biases (**Fig. 13e**) are negligible for ECMWF_IFS between 0.75 and 1 km, yet positive below and
above this altitude range. The coinciding overestimation of cloud at these heights indicates that the IFS has simulated too much
condensation, driven by the availability of too much moisture. Similarly, all UM simulations exhibit a moist bias between 0.5
and 1.6 km, between the modelled cloud layers, and exhibit small dry biases where too much cloud is modelled (e.g., 0.5 km).
These results indicate that both models have an excess of water vapour, particularly below 3 km, where negligible/dry biases
with comparison to observations are in fact an artefact of too much condensation and resulting cloud cover. This excessive
cloud cover, on the other hand, has a negative effect on the temperature bias profile, resulting in strong cold biases.
The models are in good agreement with the observed $LWC$ during period 6, with the exception of UM_CASIM-100 which
produces double the observed $LWC$ at 0.7 km (**Fig. 13g**). In particular, ECMWF_IFS performs well below 2.5 km in terms of
$LWC$, $IWC$, and cloud occurrence, with the largest difference in the latter occurring at approximately 0.7 km (100% in
ECMWF_IFS in comparison to 79% observed). Consequently, the $T$ biases are smaller during period 6 than period 3 for
ECMWF_IFS. However, these $T$ biases are still present (**Fig. 13i**), peaking at –0.96 K at 0.65 km, likely caused by this minor
overestimation in cloud cover, albeit with representative microphysics.
The magnitude of the $T$ biases for the UM simulations is similar between both periods, likely caused by this model producing
up to 100 % cloud cover at low altitude. All UM simulations exhibit stronger $T$ and $q$ biases below 1 km than ECMWF_IFS
during period 6 (**Fig. 13i—j**). Strong negative $T$ biases accompany the overestimation of cloud cover in each UM case, and the
improved model-observation agreement of $LWC$ by UM_RA2M and UM_RA2T does little to alleviate these biases with
comparison to the overestimated $LWC$ of UM_CASIM-100. Simply, there is too much low-altitude (below 1 km) cloud causing
too much cloud-top radiative cooling in the model, no matter which representation of cloud microphysics or large-scale cloud
is used.
However, while the $q$ biases were negligible when ECMWF_IFS exhibited particularly strong $T$ biases during period 3, $q$
biases for the UM become notably negative for the same effect during period 6; this is the largest dry bias simulated over the
four periods considered (with periods 4 and 5 included in the **Supporting Information**). The surface $q$ bias for the UM
simulations is smaller during period 6 than during period 3, and the tropospheric $q$ bias is positive less often, suggesting the
positive moisture bias hypothesised previously (leading to too much condensation and cloud cover) is not ubiquitous in the
model. In fact, results shown in **Fig. 13**, and **Fig. S7** for periods 4 and 5, suggest that either the increased synoptic activity or
freezing sea ice conditions (or both) of periods 5 and 6 acts to reduce this moist bias in the UM.
In summary, both models exhibit strong negative $T$ biases at altitudes coinciding with too much liquid-dominated cloud (e.g.,
**Fig. 13a, b, d**), likely caused by the consequent enhancement of cloud-top radiative cooling. $q$ biases improve where cloud is





modelled during the melt period (**Fig. 13a, e**), suggesting that the $q$ field was perhaps too moist below 3 km to begin with,
leading to too much condensation and excessive cloud cover. However, this hypothesis does not appear to be valid during the
freeze nor at altitude, as indicated by the negative $q$ biases above 2.5 km which occur where more cloud was observed than
modelled: for example, 2.5 to 4 km during period 3 for all simulations (**Fig. 13a, e**), or 2.5 to 3.5 km for the UM simulations
during period 6 (**Fig. 13f, j**). In these instances, our models produce too little cloud as they are too dry to facilitate cloud
formation. With underestimated cloud formation, the models are also slightly too warm (approximately 0.3 K) due to the
missing radiative cooling occurring at these altitudes in reality.
While the model $T$ biases align well with their overestimation of cloud cover, our analysis thus far does not account for the
height of the capping inversion. Therefore, incorrect placement of cloud in the models, or a too-deep or too-shallow modelled
BL, could be contributing to these biases and thus could affect the interpretation of our results.
**Figure 14** shows the strongest temperature inversion base identified from each model simulation and the radiosonde
measurements. In each dataset, the strongest inversion below 3 km was identified (following **Vüllers et al., 2021**); if a weaker
inversion was modelled at a lower altitude which was closer to the inversion base identified from the radiosonde, the model
inversion height was adjusted accordingly. In keeping with previous analysis, radiosonde and IFS data were interpolated to the
UM vertical grid for fair comparison; this procedure smooths some high-altitude detail in the radiosonde profiles, such that the
strength of some higher-altitude inversions is reduced causing weaker low-altitude inversions to be identified as the primary
inversion instead.
These results indicate that the strongest (unadjusted) inversion in each simulation is often too high (grey points, **Fig. 14**), and
weaker inversions at lower altitude are typically in better agreement with identified inversions from radiosondes. Low inversion
bases (below approximately 0.5 km) are consistently overestimated in each simulation, particularly during the melt period (not
shown), supporting our previous deduction that the model inversions were often too high. The detection algorithm does fail to
capture some inversions, predominantly during the freeze period, and instead underestimates the modelled inversion base
during this time window with comparison to measurements (lower right-hand points in each panel).
Modelled and observed temperature profiles were scaled using these identified inversions to remove the differences in inversion
height from our interpretation of the model biases (**Fig. 15**). When averaged over the full drift, the models are largely biased
warm below the inversion and cold above (up to 3 km; **Fig. 15a**), with the exception of UM_CASIM-100 which also exhibits
a subtle cold bias just below the inversion. This warm below/cold above signal is more consistent between the models during
the melt period (**Fig. 15b**). Above the inversion, ECMWF_IFS exhibits a stronger cold bias than the UM simulations. The
shape of the scaled profile is rather consistent between the melt and freeze with ECMWF_IFS; the model is consistently too
warm below the inversion, and too cold above, with comparison to radiosonde measurements. However, the UM simulations,
particularly UM_RA2M, are partially biased cold below the inversion during the freeze. As previously mentioned, biases during
the freeze period must be interpreted with caution as the inversion detection algorithm performed less well during this time
window, with several modelled inversions missed. However, these scaled $T$ bias profiles support our previous hypothesis that
cloud longwave cooling is producing colder thermodynamic conditions in the models than were observed, irrespective of the





differences between modelled and observed inversion heights. Similarly, the warm surface bias indicated previously can be
interpreted to span most of the lower troposphere below the main inversion base, rather than solely near the surface.
**4      Discussion**
**4.1     Surface radiative balance**
**4.1.1    Shortwave**
The small $SW_\downarrow$ biases exhibited by the standard UM configurations concurrent with a more significant $SW_{net}$ bias indicate that
the modelled surface albedo is likely too low. While the observed albedo may be biased high due to its calculation from a
spatially small sample of sea ice (directly surrounding the ship), the UM surface albedo parameterisation has previously been
shown to be too low in the high Arctic (**Birch et al., 2009; 2012**). The temperature and albedo limits used in the standard
Regional Atmosphere parameterisation have been increased since **Birch et al. (2009, 2012)**; however, it is clear from **Fig. 2**
that the snow-on-sea-ice parameterisation limits tested here with ECMWF_IFS and UM_CASIM-100 (currently used in the
Global Atmosphere model configuration) produce a better comparison with our high Arctic measurements.
**4.1.2    Longwave**
The root of the $LW$ error in each simulation is likely the >90% liquid-dominated low cloud occurrence which is not
representative of the observations (**Fig. 4b**). This problem has been previously identified in the high Arctic with both models
used in this study (**Birch et al., 2012**; **Sotiropoulou et al., 2016**) and recent model improvements/microphysical changes have
not sufficiently improved model performance in this regard. The positive $LW$ biases are consistent with the too-warm surface
$T$ biases in all simulations (**Figs. 9, 10**), which is also consistent with previous findings with both models (**Birch et al., 2009**;
**Sotiropoulou et al., 2016**) and with the ERA-Interim reanalysis product (**Jakobson et al., 2012**; **Wesslén et al., 2014**). **Figures**
**9** and **10** suggest that the UM simulations are perhaps better at capturing the near-surface $T$ over the freeze, while ECMWF_IFS
consistently has a warm surface bias regardless of season. **Tjernström et al. (2021)** suggest that surface is actually warmed by
the atmosphere in the IFS, not the opposite, as indicated by the enhanced downward sensible heat flux, in combination with
diminished $SW_\downarrow$ with comparison to observations.
Given these results, we suggest that excessive cloudiness is likely a contributing factor to the warm surface bias in all
simulations. In particular, it is noteworthy that UM_CASIM-100 performs most poorly of the UM simulations. This result is
disappointing given the improvement of UM_CASIM-100 over the standard Regional Atmosphere configurations in both $SW_\downarrow$
and $SW_{net}$. Including CASIM aerosol processing through wet scavenging – thus enabling cloud dissipation (e.g., **Stevens et**
**al., 2017**) – may rectify this issue, or the representation of prognostic ice nucleating particles in place of a simple diagnostic
relationship between temperature and cloud ice number concentrations (e.g., **Varma et al., 2021**). These pathways will be
explored in future work; however, it is highly likely that other meteorological factors and incorrect model processes are
contributing to this warm surface bias across all of our simulations, in addition to cloudiness.



## 4.2 Lower troposphere

### 4.2.1 Temperature

Temperature biases are strongest within the lowest 3 km of our model domains (**Figs. 9, 13**); this is also the altitude range over which the models overestimate cloud occurrence. With too much cloud, cloud top radiative cooling likely lowers the temperature too efficiently; this, coupled with incorrect cloud positioning (e.g., period 3; **Fig 13**), gives a cold bias with respect to our observations, above the observed main capping inversion. Where the liquid (and ice) phase is modelled more effectively – e.g., ECMWF_IFS during period 6 (**Fig. 13**) – the associated median biases are smaller ($< \pm 1$ K), supporting this conclusion.

The dipole in $T$ errors shown in **Figs. 8** and **9**, with a positive bias towards the surface below a negative bias between 0.5 and 3 km, suggests that heat and moisture are not being sufficiently transported upwards from the surface or downwards from cooling at cloud top. This $T$ bias is present in all simulations during both the melt and freeze periods and could result from the models failing to reproduce the structure of more than one strong observed inversion, instead exhibiting comparatively smooth $T$ profiles. As shown by **Fig. 14**, low-altitude $T$ inversions are often overestimated by the models, particularly during the melt period (not shown): this too-deep surface mixed layer likely results in incorrect cloud placement, leading to thermodynamic model-observation biases on a 1-to-1 comparison. Our scaled thermodynamic analysis (**Figs. 14, 15**) indicates that, while the models are often incorrectly placing the temperature inversion (consistent with previous findings; **Birch et al., 2012**), the relationship between too much cloud and strong negative $T$ biases suggested by **Fig. 13** appears to be robust under these scaled height adjustments by the inversion base, and that the simulations are still largely too warm below the inversion.

Both **Sotiropoulou et al. (2016)** and **Tjernström et al. (2021)** found a similar vertical structure of the temperature biases with the IFS model, with positive biases within the lower 0.5 km of the atmosphere and a consistent cold bias present around 1 km. **Tjernström et al. (2021)** found that this cold bias intensifies with time during 3-day forecasts, indicating that it is made worse by processes within the model. They hypothesised the mid-level convection parameterisation triggering too-efficiently within the IFS could be transporting water vapour out of the BL, resulting in too much condensation to form cloud. While our UM LAM simulations do not employ such a convection scheme, the global driving model does: this parameterisation acts in addition to shallow convection, e.g., representing convection in mid-latitude storms, or above the BL where the surface layer is stable. Given the apparent close relationship between the biases exhibited by the LAM and global model **(Fig. 9, 11),** we conducted a short 6-day test with the global mid-level convection scheme switched off; this test caused over an order of magnitude increase in the UM_GLM cold biases shown in **Fig. 11** (not shown). Given the short duration and extreme "on-off" nature of this test, these results do not provide conclusive evidence that the mid-level convection scheme is not contributing to the thermodynamic biases shown here. Further investigation into vertical transport and mixing of scalars (temperature, moisture and clouds) is needed to confirm the origin of these thermodynamic biases. Specifically, more investigation into vertical transport and mixing of scalars (temperature, moisture and clouds) is needed; however, this investigation is beyond the scope of this paper.

### 4.2.2 Moisture

During the melt period, our results indicate that the UM is particularly moist throughout much of the troposphere (**Fig. 9b**), suggesting that the melting ice is enabling a too-great moisture source from the surface to the atmosphere. However, this





tropospheric bias appears to be rectified during the freeze, while the surface bias remains (**Fig. 9d**); therefore, the hypothesised
melting ice source is likely not the only contributor of this moisture bias. Latent heat fluxes measured during the expedition
indicate no significant change between the melt and freeze periods (not shown); therefore, the hypothesised increased moisture
flux during the melt is unlikely.
Given the close relationship between our UM LAM and global model biases (**Figs. 9, 11**), increased poleward moisture
transport introduced at the lateral boundary conditions from the mid-latitudes could partly explain these biases. This
phenomenon has been previously identified to be a consequence of climate change and may promote increased cloudiness in
the polar regions (e.g., **Held and Soden, 2006**; **Vavrus et al., 2009**; **Allen et al., 2012**; **Bender et al., 2012**). The moist surface
bias is also present over both the melt and freeze in ECMWF_IFS; however, ECMWF_IFS is routinely too dry between 0.5
and 4 km, in contrast to the UM. Instead, the IFS traps too much moisture in the lowest 0.5 km, suggesting that the upward
transport of moisture may be insufficient, the cloud sink above 0.5 km is too great, or there are consistent biases introduced via
assimilation of data other than the radiosonde data (e.g., satellite).
The moist bias exhibited by ECMWF_IFS towards the surface has previously been highlighted by **Sotiropoulou et al. (2016)**,
who suggested that this problem may explain why this model struggles to reproduce humidity inversions above the BL. There
are instances where negative $T$ biases coincide with negative $q$ biases at altitudes just above the main temperature inversion
(for example, at 27 Aug; **Fig. 9**). Moisture inversions have often been observed during the Arctic summertime (**Sedlar et al.,**
**2012; Nygård et al., 2014**); ECMWF_IFS fails to reproduce such inversions observed during AO2018. This dry bias above
the observed capping inversion around 27 Aug is not as strong in the UM simulations, but the UM does successfully reproduce
a small humidity inversion.
**4.3     Cloud macro- and microphysics**
The UM simulations have >98% cloud occurrence around 0.2 km over all four periods. Reduced $SW_\downarrow$ biases with respect to the
standard Regional Atmosphere configurations indicate that UM_CASIM-100 does improve agreement with our high Arctic
observations (**Table 4**). In particular, the comparison between UM_CASIM-100 and UM_RA2M (which use the same large-
scale cloud scheme and differ only in their representation of resolved cloud physics) shows that the new CASIM scheme
reproduces the observed Arctic clouds better on a microphysical level.
The ice phase differs more between the models than the liquid phase, likely due to its strong relationship with temperature:
UM_RA2M and UM_RA2T use the **Fletcher (1962)** parameterisation for primary ice formation, while ECMWF_IFS uses
**Meyers et al. (1992)** and UM_CASIM-100 uses **Cooper (1986)**. Each of these parameterisations is inherently temperature-
dependent, with **Meyers et al., (1992)** producing the largest ice number concentration, and **Fletcher (1962)** producing the
smallest. Given that each simulation does not reproduce the observed temperature profile well below 3 km, the onset of ice
nucleation (occurring below a threshold of –10 °C in the UM, for example) will be affected. If ice production is triggered
prematurely, cloud liquid properties should be dampened via the Wegener-Bergeron-Findeisen mechanism; evidence of this
can be seen in UM_RA2T during period 3, where an overestimation of ice below 2 km corresponds with a smaller mean LWC
than the other simulations (**Fig. 13b—c**).





When considering the drift as a whole, $IWC$ is overestimated by all simulations (except UM_RA2M) below 1.5 km, where our
T and q biases are at their greatest. To test whether the method of parameterising primary ice itself has any effect on these
biases, we used the **Fletcher (1962)**, **Cooper (1986)**, and **Meyers et al. (1992)** parameterisations over a short test period within
the CASIM framework; however, we found little difference in the tropospheric ice with the different parameterisation methods
(**Fig. S8**). Given the spread in $IWC$ results shown here, further investigation into the best methods to represent primary ice
production in such global and NWP models should be considered in future, with specific focus on employing prognostic ice
nucleating particles (similar to the CCN used here in UM_CASIM-100) to facilitate ice formation rather than simply using a
temperature-dependent function (e.g., **Varma et al., 2021**). The primary ice parameterisations used here do affect the clouds
modelled – for example, with a large spread in modelled $IWC$ aloft during period 3 (**Fig. 13c**) – and a more realistic
representation of the ice phase would likely contribute to improved cloud liquid properties. In particular, it is likely that ice
formation at warm supercooled temperatures (> –10 °C) will be of importance given the overestimated dominance of cloud
liquid at low altitude in our simulations.
Below 3 km, the mean modelled $LWC$ often exceeds the observed value, with better agreement between 0.15 and 1 km than
between 1 and 3 km (**Fig. 4, 13**). This overestimation of cloud liquid is also evident from the $LWP$ data, with each simulation
exhibiting a greater $LWP$ than was measured (**Fig. 5**) when averaged over our meteorological periods (not shown). At first
glance, the simulations agree reasonably well with measurements (**Fig. 5a**), but this subtle overestimation is clear from the
PDFs (**Fig. 5b**). The exception to this is UM_RA2T; this is the only simulation which often underestimates $LWP$, due to its
increased cloud ice mass within the lower troposphere in comparison to the other simulations (**Fig. 4d**). For example, the mean
measured $LWP$ during period 3 is 122.8 g m$^{-2}$, yet UM_RA2T only produces 70.4 g m$^{-2}$. In contrast, UM_RA2T reproduces
the mean measured $LWP$ well during period 6 (48.5 g m$^{-2}$ measured versus 43.2 g m$^{-2}$ modelled), with agreement improving
with time throughout the drift. This efficient ice-producing simulation suggests that the ice phase influences cloud properties
as time progresses more so in reality, while the other UM cases, with less dominant ice, retain too much liquid in comparison
to the measurements. To an extent, ECMWF_IFS also behaves in this way, retaining too much cloud liquid; however, it
performs much better than UM_RA2M and UM_CASIM-100 in reproducing the mean $LWC$ and $IWC$ during period 6 (**Fig.
13g, h**).
These simulations suggest that the model development community has effectively gone too far with the reduction of the ice
phase in central Arctic mixed-phase clouds. The surface $LW$ balance is positively biased, and these excessive low-level clouds
are a contributing factor: by enabling too much liquid to form, and restricting the ice too efficiently, these clouds efficiently
absorb and re-emit upwelling $LW$ radiation back towards the surface. Our results show that we have made great improvements
in the $SW$, driven by the improvements we have made to our cloud physics representation in these models (in addition to a
better estimation of the surface albedo). However, the too-consistent cloud cover coupled with too much cloud liquid is
hampering our model capability, and further developments (such as the inclusion of representative CCN and INP inputs to
double-moment cloud schemes to facilitate cloud dissipation) will likely go some way to tackle this issue.



## 5    Conclusions


Model simulations with the Met Office UM and the ECMWF IFS were compared with observations made during the Arctic
Ocean 2018 expedition to evaluate model performance in the high Arctic, with particular focus on modelled clouds and the
surface radiative balance. Four key simulations were considered: a global configuration with the IFS and three nested
configurations with the UM (each using different combinations of large-scale cloud and microphysics schemes yet driven by
the same global model setup). These four simulations were compared with observations by using Cloudnet to build model-
comparable cloud fractions and water contents and thus identify consistent process weaknesses between the model
configurations chosen.
We found that key issues identified in previous studies, such as positive near-surface temperature biases (**Sotiropoulou et al.,**
**2016**), remain problems in recent releases of both the UM and IFS. Modelled BLs are often too deep (**Fig. 9, 14**), particularly
during the melt period, and thermodynamic biases, cloud occurrence, and cloud microphysics are consequently in poor
agreement with observations below 3 km. Excessive low-cloud occurrence is prevalent in both models (**Fig. 3**) and no
simulation adequately reproduces cloud-free periods and associated increases in longwave net emission (**Fig. 2**), consistent
with previous UM and IFS evaluations in the Arctic (**Birch et al., 2012**; **Sotiropoulou et al., 2016**). Strong negative
temperature biases (**Figs. 8, 9, 13**) coincide with too-frequent liquid-dominated cloud layers (**Fig. 13a, b, f, g**), likely associated
with over productive cloud-top radiative cooling in the models. Cloud liquid and ice water contents, especially below 1 km,
were within an order of magnitude of our observations (**Fig. 4**), but clouds occurred too frequently, contained too much liquid
between 1 and 3 km, and were often at too-high an altitude (**Fig. 13a—c**).
Radiative interactions are in better agreement with observations and all models capture the observed distribution of $SW_{net}$ and
$LW_{net}$ better during the sea ice freeze period in comparison to the melt period (**Fig. 9**). Improved radiative interactions and
thermodynamic biases during the freeze can be linked with improved agreement of cloud occurrence and microphysics **(Fig.**
**13, S7).** We found that the surface albedo in each model configuration is underestimated with respect to observational estimates
(see **Supporting Information**), but this is unsurprising given the models are representing an average albedo over a 1.5 / 9 km
grid box while our observed estimates are from the area immediately surrounding the ship. Updating the surface albedo
parameterisation limits used within the UM Regional Atmosphere configurations (UM_RA2M/UM_RA2T) to those used in
the Global Atmosphere GA6.0/6.1 configuration (UM_CASIM-100) greatly improves our surface albedo comparison with
observational estimates (see **Supporting Information**) and thus contributes to the good comparison of UM_CASIM-100 with
measured shortwave radiation data.
We propose that four factors are important to failings in our model simulations:
1.   The choice and use of large-scale cloud schemes at high resolution:
o    Both the UM and IFS poorly capture Cloudnet-calculated cloud fractions from observations over Aug—Sep
2018 in the central Arctic, particularly at altitudes between 2 km and 8 km (**Fig. 4a**). Building a comparable
mask based on $TWC$ shows that the cloud modelled aloft is actually in good agreement with observations
(**Fig. 4b**), while highlighting that the over prediction of cloud occurrence below 3 km is in fact much worse
than suggested by the $C_V$ comparison. As such, we suggest that cloud fractions should not be used in isolation





as a model comparison metric over the Arctic as models represent this parameter differently at the present

time (as detailed in the **Supporting Information**).

2. The cloud microphysics scheme chosen to represent resolved clouds:

○  UM_CASIM-100 performs best in terms of $SW_{net}$ (**Fig. 2**, **Table 4**), but it struggles to capture cloud-free

episodes, thus producing a $LW_{net}$ PDF which is too narrow in comparison to our measurements.

○  ECMWF_IFS shares the too-narrow $LW_{net}$ PDF of UM_CASIM-100; however, it often produces a IWC in

reasonable agreement with observations, and its mean $LWC$ profile does agree particularly well with the

observations at times (e.g., period 6; **Fig. 13g**).

○  Of the UM simulations considered, UM_CASIM-100 is in best agreement with both ECMWF_IFS and

observations in terms of net radiation, $SW_{net}$ and $SW_{\downarrow}$. This improved radiative agreement can be linked to

its better cloud microphysical agreement with our Cloudnet-derived cloud liquid water content over the

standard Regional Atmosphere configurations (**Figs. 4, 5, 13**); however, UM_CASIM-100 produces even

poorer cloud fractions aloft than either UM_RA2M or UM_RA2T.

3. Representative CCN concentrations, and thus droplet number concentrations, as a function of altitude:

○  Representative CCN concentrations in UM_CASIM-AeroProf somewhat improves the overestimation of $q_{liq}$

within low level clouds in UM_CASIM-100. However, the $q_{liq}$ decrease is not sufficient to trigger an

increase in liquid precipitation, which would thus decrease cloud lifetime, so the modelled $C_V$ is essentially

unchanged (**Fig. 7**). Crucially, thermodynamic biases with respect to observations are not improved through

this enhanced complexity (**Fig. 10**), highlighting that these biases may not be fixed by a more comprehensive

representation of cloud physics. Further work is required, with the inclusion of wet scavenging of aerosols

and prognostic INP, to rule out whether such processes could improve the model biases over and above the

inclusion of representative aerosol concentrations alone.

4. The global model analyses used to produce boundary conditions for high resolution nests:

○  The thermodynamic biases identified in our models differ only a little between the UM simulations despite

differences in their cloud configurations. Comparisons with the global model show that the biases within the

LAM are largely inherited from the global model and associated DA system (**Fig. 9, 11**); therefore, for LAM

configurations such as that tested here, we will not obtain the true benefit of more sophisticated cloud

microphysics schemes in NWP simulations until we address the large-scale biases in their driving models/DA

system.

While representative CCN concentrations are indeed important for properly reproducing Arctic cloud structure and its
consequential impact on the net surface radiation, our findings indicate that such representative cloud nuclei inputs still have
little impact on thermodynamic biases in the lower troposphere. For our given LAM configuration, we speculate that these
biases will always be inherited from the driving model/DA and will continually bias cloud formation processes and BL depth;
however, using an increased domain size, with the science area of interest as far from the lateral boundaries as possible, may
help to reduce the influence of the driving model/DA. The issue of inherited thermodynamic biases is concerning as both the
UM global model and IFS are both used within the community to drive NWP configurations of the same model (UM) or others





(IFS). For example, the IFS configuration tested here is similar to that used by ERA5; therefore, these biases could influence
future high Arctic NWP simulations if these reanalyses are used for initialisation.
Our recommendations are thus twofold. To improve our Arctic cloud modelling capability, we must continue to improve the
cloud physics description striving for an optimum complexity, such as the introduction of representative CCN concentrations
and double-moment cloud liquid illustrated here, in addition to the inclusion of prognostic INP and associated aerosol
processing mechanisms. However, we must concurrently address the overabundant occurrence of a too-well-mixed and too-
cloudy lower troposphere, and tackle the resultant thermodynamic biases, in our global driving models and their respective DA
systems.





#### Appendix A    Cloudnet mishandling of fog data


$LWP$ measurements from the HATPRO microwave radiometer were used in this study; this instrument provides measurements
of microwave brightness temperatures, from which $LWP$ is derived for the full atmospheric column above the instrument (here
located approximately 13 m above the surface). This includes measurement of liquid clouds at altitudes below the radar's first
range gate at 156 m. Fog periods occurred frequently throughout the expedition (**Vüllers et al., 2021**); therefore, we had several
instances where liquid fog was measured with the HATPRO with small quantities of liquid, or none, detected in the clouds
above (from lidar/radar).
Cloudnet calculates an offset to be deducted from the $LWP$ time series dependent upon its categorisations of cloud to ensure
that liquid is partitioned throughout the cloud column only if liquid clouds were present. This offset is non-uniform, calculated
as a given fraction of the $LWP$ signal on a daily basis, and is deducted from the $LWP$ data to ensure liquid partitioning is
conducted correctly within the Cloudnet algorithm. Given the frequency of fog occurrence, this offset was often overestimated
and too much liquid data were removed, thus negatively impacting the $LWP$ and $LWC$ comparisons with our model simulations.
To rectify this problem, we removed the $LWP$ offset calculation from the Cloudnet procedure, enabling all ingested data to be
used by Cloudnet. We then compared these adapted Cloudnet $LWC$ data to a $LWC$ calculated under an adiabatic assumption to
test whether the latter could be used as an approximation of the true $LWC$ if there was not as much fog present during the
expedition. **Figure A1** shows this comparison using all data from the drift period and indicates that, by keeping all fog liquid
data in the time series, Cloudnet artificially partitions these data to liquid cloud layers identified by the lidar, leading to too
much liquid in clouds within the lowest 1 km of the atmosphere (with comparison to the adiabatic profile). These data also
indicate that we can safely use the adiabatic $LWC$ as this artificial liquid enhancement is confined to the lowest 1 km and does
not significantly affect the comparison for higher altitudes. Following these comparisons, we chose to include the adiabatic
$LWC$ in our comparisons with model simulations to exclude the artificial enhancement of cloud liquid at low altitudes in our
measurement data.





**Tables**

| Table 1: List of key abbreviations, acronyms, and parameters referenced in this study. | |
|---|---|
| **Label used** | **Description** |
| $\beta$ | Lidar backscatter |
| BL | Boundary layer |
| $C_V$ | Cloud fraction by volume, determined using Cloudnet |
| CASIM | Cloud-AeroSol Interacting Microphysics |
| CCN | Cloud condensation nuclei |
| ECMWF | European Centre for Medium-Range Weather Forecasting |
| GCM | General Circulation Model |
| IFS | Integrated Forecasting System |
| $IWC$ | Cloud ice water content |
| LAM | Limited Area Model |
| $LW_{net}$ | Net longwave radiation at surface |
| $LW_\downarrow$ | Downwelling longwave radiation at surface |
| $LWC$ | Cloud liquid water content |
| $LWP$ | Liquid water path |
| $N_d$ | Cloud droplet number concentration |
| $q$ | Specific humidity |
| $q_{ice}$ | Cloud ice mixing ratio |
| $q_{liq}$ | Cloud liquid mixing ratio |
| $R_{net}$ | Net total radiation at surface |
| $RH$ | Relative humidity |
| $RH_{crit}$ | Critical relative humidity for condensation in models |
| $SW_{net}$ | Net shortwave radiation at surface |
| $SW_\downarrow$ | Downwelling shortwave radiation at surface |
| $\theta_e$ | Equivalent potential temperature |
| $T$ | Temperature |
| $TWC$ | Total cloud water content |
| UM | Unified Model |
| $Z_e$ | Radar reflectivity |






| Simulation | Details | References |
|---|---|---|
| ECMWF_IFS | Cy46r1; cloud and large-scale precipitation following update to Cy36r4. Snow included in all cloud fraction and cloud ice water content analyses. | **Forbes and Ahlgrimm (2014)** |
| UM_CASIM-100 | UM with LAM using CASIM scheme operating with 100 cm$^{-3}$ accumulation mode aerosol particles over the full model column and across the entire LAM. Droplet activation: **Abdul-Razzak and Ghan (2000)**; primary ice formation: **Cooper (1986)**. Diagnostic cloud fraction and condensate from large-scale cloud (**Smith, 1990**) scheme. | **Smith (1990)**; **Hill et al. (2015)**; **Grosvenor et al., (2017)**; **Kupiszewski et al. (2013)** |
| UM_RA2T | UM with LAM operating with the tropical regional atmosphere configuration (RA2T). Prognostic cloud and prognostic condensate (PC2) cloud scheme used with cloud microphysics based on **Wilson and Ballard (1999)**. | **Wilson and Ballard (1999)**; **Wilson et al. (2008)**; **Bush et al. (2020)** |
| UM_RA2M | UM with LAM operating with the mid-latitude regional atmosphere configuration (RA2M). **Wilson and Ballard (1999)** cloud microphysics scheme with diagnostic cloud fraction and condensate from large-scale cloud (**Smith, 1990**) scheme. | **Smith (1990)**; **Wilson and Ballard (1999)**; **Bush et al. (2020)** |
| UM_GLM | UM global model operating a N768 resolution (corresponding to approximately 17 km at the mid-latitudes) using the Global Atmosphere 6.1 configuration with a rotated pole. Uses 70 quadratically-spaced vertical levels up to 80 km with PC2 large-scale cloud and cloud microphysics based on **Wilson and Ballard (1999)**. Data over the full drift period are included to contextualise thermodynamic profiles extracted from the UM LAMs. | **Walters et al. (2017)**; **Wilson et al. (2008)**; **Wilson and Ballard (1999)** |
| UM_CASIM-AeroProf | As UM_CASIM-100, except day-averaged soluble coarse- and accumulation-mode concentrations from UKCA are input in place of the constant profile (see **Supporting Information** for details) to indicate role of realistic aerosol number concentrations. | **Morgenstern et al. (2009)**; **O'Connor et al. (2014)**; **Mann et al. (2010)** |

Table 2: Summary of the four model configurations to simulate cloud and thermodynamic conditions observed over the full AO2018 drift period in this study. Three additional simulations included for further investigation of results are listed in shaded sections below.



| UM_RA2M-ERAI-GLM | As UM_RA2M LAM configuration, except using ERA-Interim data to initialise the UM global model instead of standard global start dumps. Data only included from a short subset of the drift period (31 Aug to 5 Sep) for further analysis of temperature and moisture profiles. | **Dee et al. (2011)** |
|---|---|---|




Table 3: Summary of cloud microphysical process representation in each simulation setup. Chosen processes are highlighted as key differences between the schemes. k = model level; Z = altitude.

| | Model simulation | | | |
|---|---|---|---|---|
| **Physical process** | **ECMWF_IFS** | **UM_CASIM-100** | **UM_RA2T** | **UM_RA2M** |
| Prognostic cloud variables | Cloud fraction, vapour, cloud liquid, cloud ice, rain, and snow (single moment) | Vapour, cloud liquid, cloud ice, graupel, rain, and snow mixing ratios and number concentrations (double moment) | Liquid, ice, and total cloud fractions; vapour, cloud liquid, cloud ice (all ice, includes snow), and rain (single moment). | Vapour, cloud liquid, cloud ice (all ice, includes snow), and rain (single moment). |
| Large-scale cloud fraction (*described in*) | Prognostic (**Tiedtke, 1993**) | Diagnostic (**Smith, 1990**) | Prognostic (**Wilson et al., 2008**) | Diagnostic (**Smith, 1990**) |
| Droplet number concentration | Diagnostic. Wind-speed dependent function for radiation calculations (following **Martin et al., 1994**). For auto-conversion, diagnosed by land-surface mask (ocean surface, fixed); 50cm$^{-3}$. | Prognostic; **Abdul-Razzak and Ghan (2000)**, referencing an accumulation mode aerosol profile of 100cm$^{-3}$ at all Z. | Diagnosed by land-surface mask (ocean surface, fixed); 100 cm$^{-3}$. Tapered to 50 cm$^{-3}$ at $Z \leq 50$ m from 150 m. | Diagnosed by land-surface mask (ocean surface, fixed); 100 cm$^{-3}$. Tapered to 50 cm$^{-3}$ at $Z \leq 50$ m from 150 m. |
| Critical grid-box mean RH for condensation | $RH_{crit} = 0.8$, increasing towards the BL as a function of height. | 0.96 at the surface and decreases monotonically upwards to 0.80 at 0.85 km, above which it remains constant with altitude (k >= 15) (**Grosvenor et al., 2017**) | 0.96 at the surface and decreases monotonically upwards to 0.80 at 0.85 km, above which it remains constant with altitude (k >= 15) (**Grosvenor et al., 2017**) | 0.96 at the surface and decreases monotonically upwards to 0.80 at 0.85 km, above which it remains constant with altitude (k >= 15) (**Grosvenor et al., 2017**) |





**Table 4**: Mean surface radiation biases (model-observations) over periods 3—6, with mean measured values for reference. Observations included are hourly-integrated values for consistency with the models. All values are in W m$^{-2}$. Smallest biases are highlighted in bold.

| Component | | Observations | ECMWF_IFS | UM_CASIM-100 | UM_RA2T | UM_RA2M |
|---|---|---|---|---|---|---|
| $SW_{net}$ | P3 | 24.35 | 16.41 | **6.09** | 39.69 | 38.02 |
| | P4 | 19.45 | 4.23 | **0.85** | 24.74 | 20.77 |
| | P5 | 9.87 | 9.75 | **7.88** | 19.81 | 18.09 |
| | P6 | 7.37 | 6.75 | **5.11** | 16.67 | 15.3 |
| $SW_{\downarrow}$ | P3 | 117.39 | -20.4 | -10.93 | 7.44 | **4.08** |
| | P4 | 72.86 | -6.84 | **-1.49** | 14.68 | 6.99 |
| | P5 | 55.55 | **0.9** | 9.7 | 16.43 | 12.66 |
| | P6 | 41.66 | **-0.3** | -2.21 | 11.86 | 7.48 |
| $LW_{net}$ | P3 | -21.38 | **9.56** | 16.32 | 10.48 | 10.71 |
| | P4 | -9.48 | 3.16 | 4.85 | **2.71** | 3.25 |
| | P5 | -11.77 | -2.96 | **1.89** | -3.77 | -3.65 |
| | P6 | -13.22 | **-0.27** | 5.46 | -3.6 | -0.29 |
| $LW_{\downarrow}$ | P3 | 285.82 | 19.0 | 22.41 | **18.42** | 18.66 |
| | P4 | 303.12 | 6.0 | 6.5 | **5.04** | 5.54 |
| | P5 | 291.53 | **0.31** | 2.6 | -2.25 | -2.15 |
| | P6 | 286.88 | 5.02 | 9.6 | **0.28** | 5.13 |






**Figures**

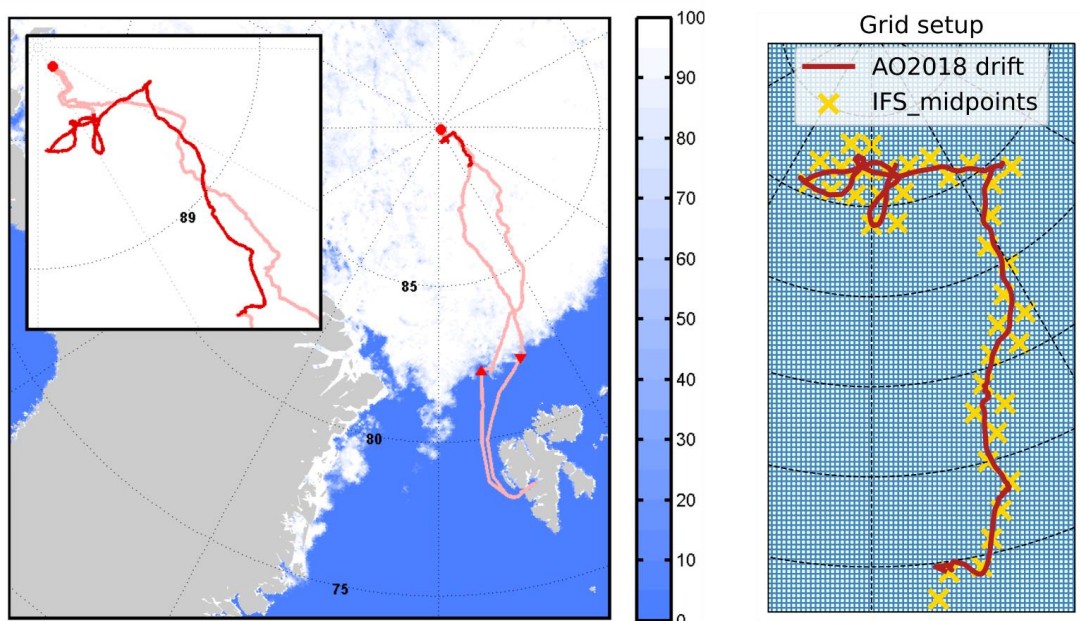

**Figure 1**: LHS: Map of cruise track and sea ice cover during AO2018 expedition from **Vüllers et al. (2021)**, with drift period (red) in inset. RHS: Ship position during the drift period (red), with grid outline for UM_CASIM-100, UM_RA2T, and UM_RA2M shown in blue and mid-points of ECMWF_IFS grid indicated by yellow crosses. Note grid size difference for illustrative purposes and not to scale: UM grid boxes are 1.5 × 1.5 km, IFS grid boxes are 9 × 9 km in size.




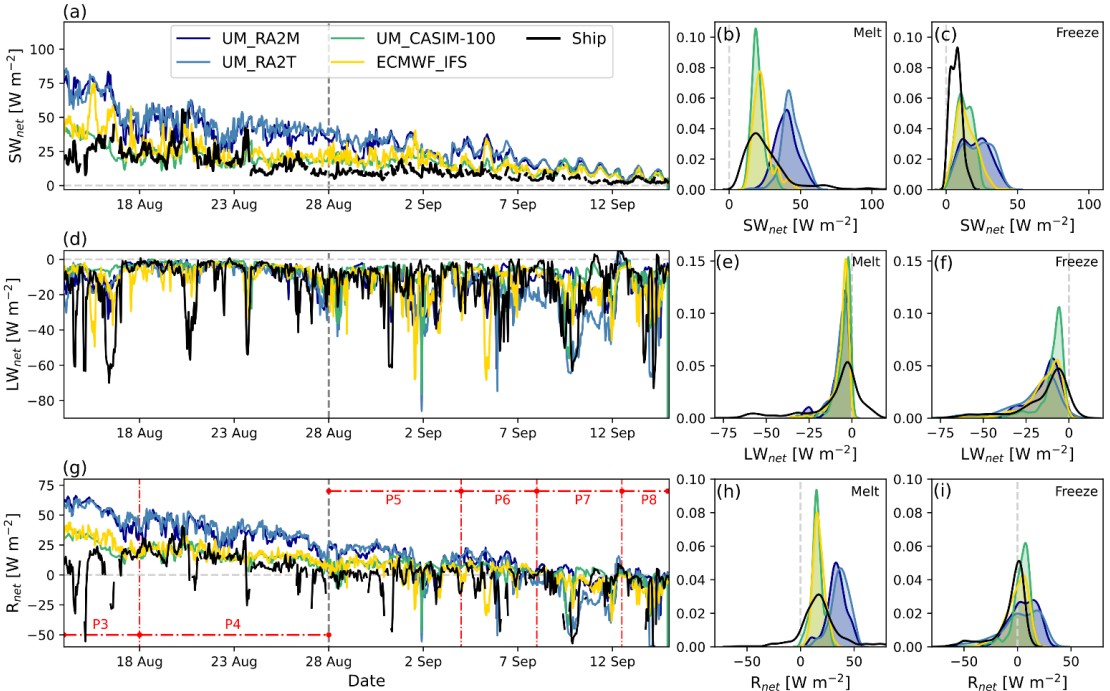

**Figure 2:** $SW_{net}$, $LW_{net}$, and $R_{net}$ simulated by UM_RA2M (dark blue), UM_RA2T (light blue), UM_CASIM-100 (green), and ECMWF_IFS (yellow). Hourly-averaged measurements on board the ship (black) shown for comparison. LHS: timeseries; RHS: PDFs. PDFs are split between melting and freezing sea ice conditions using a threshold of 28 Aug as indicated by the grey vertical dashed line in panels (a), (d), and (g). Radiation terms are defined as positive downwards. Sub-periods used in subsequent sections are marked (red) in panel (g).




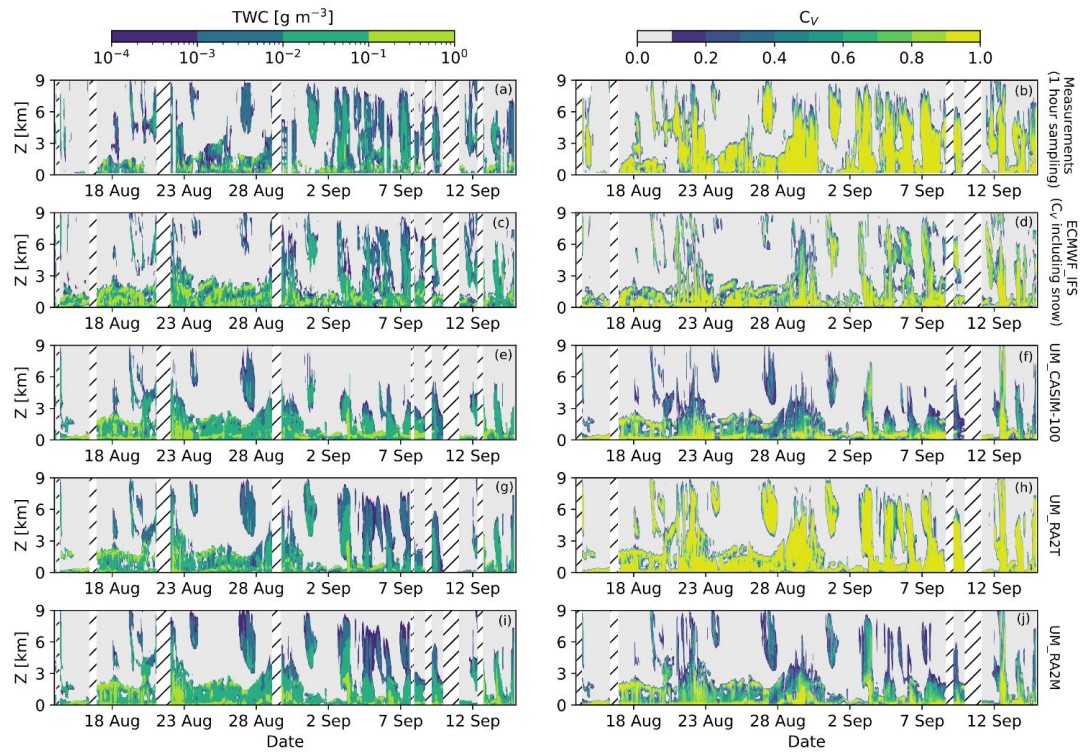

**Figure 3**: Total water content (*left, TWC*) and cloud fraction (*right, $C_V$*). (**a—b**) calculated from observations using Cloudnet and diagnosed by (**c—d**) ECMWF_IFS, (**e—f**) UM_CASIM-100, (**g—h**) UM_RA2T, and (**i—j**) UM_RA2M. Missing measurement data are indicated by hatched areas; times where data are missing from the observations are removed from the model data to provide a fair comparison. Missing data periods differ between the *TWC* and *$C_V$* products due to the different instrumentation requirements within Cloudnet for each.




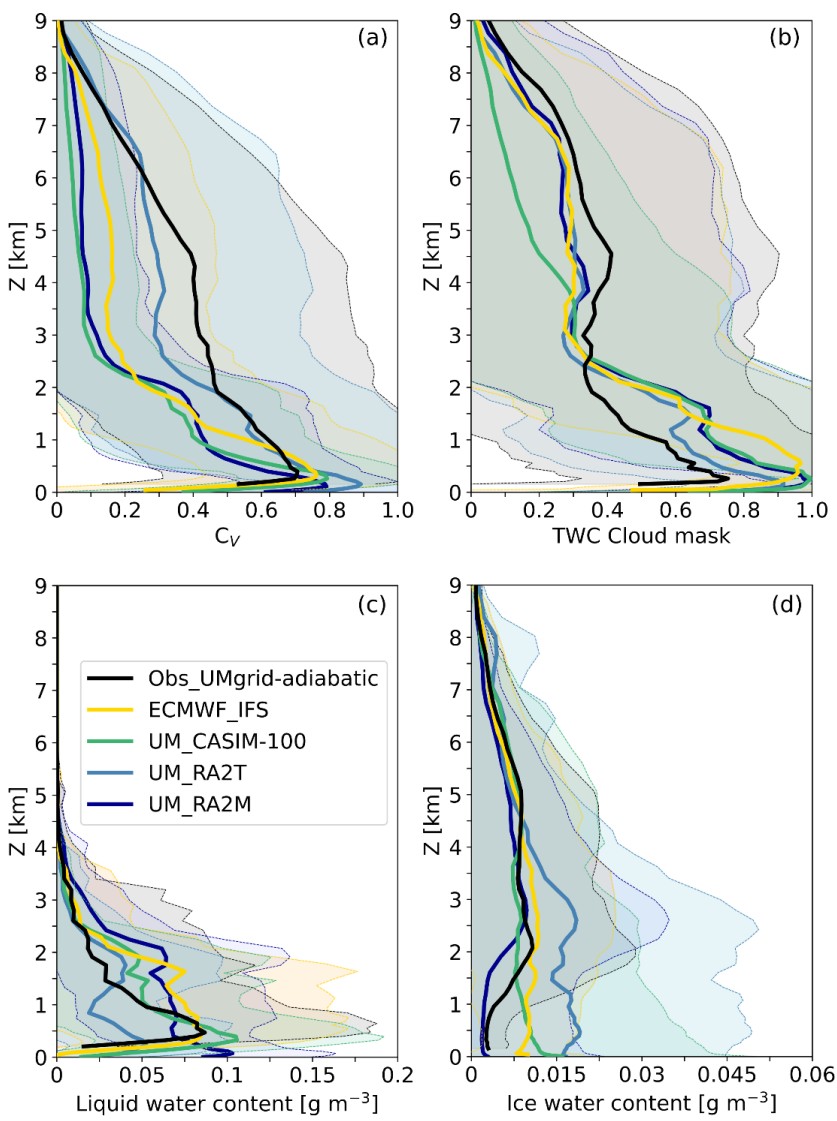

**Figure 4**: Comparison between (**a**) mean $C_V$ observed (black, calculated using Cloudnet) and modelled (UM_RA2M = dark blue; ECMWF_IFS = yellow; UM_CASIM-100 = green; UM_RA2T = light blue) over the AO2018 drift period. (**b**) $TWC$ cloud mask comparison, where masks are calculated using only in-cloud data as described in **Sect. 2.4**. (**c—d**) Same comparison for liquid and ice cloud water contents respectively, using in-cloud data only. $LWC$ data from the observations are calculated using Cloudnet by assuming an adiabatic profile (see **Appendix A**). Lines indicate the mean profiles of each dataset, shaded areas depict ± one standard deviation from the mean. Uncertainties associated with the retrieval process are not shown.


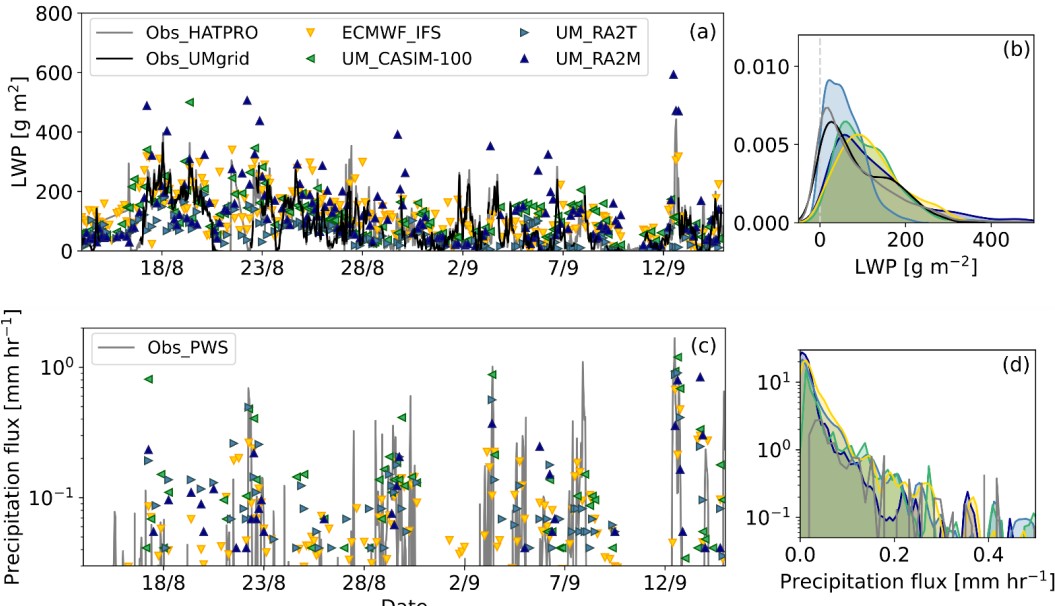

**Figure 5**: Timeseries of (**a—b**) liquid water path (*LWP*) and (**c—d**) total precipitation flux at the surface over the full drift period. (**a—b**) HATPRO measurements (grey) are included for comparison with the model data (coloured markers). *LWP* data averaged on to the UM grid by Cloudnet are shown in black (*Obs_UMgrid*). (**c—d**) Weather sensor (PWS) measurements of total precipitation from the 7th deck (grey) are included for comparison with model rain and snow fields. (**a, c**) model data shown every 3 hours for clarity; (**b, d**) all model data included for comparison.





**Figure 6**: Model biases in radiation terms ($SW_\downarrow$ (*left*), $LW_\downarrow$ (*middle*), and $R_{net}$ (*right*)), *LWP*, and $C_V$. Model-observation biases are calculated hourly for the radiation and *LWP* terms using measurements from the ship-based radiometers and HATPRO microwave radiometer, respectively. Shading: model-observation difference between mean $C_V$ below 3 km, where model data below the height of the lowest radar range gate (156 m) is excluded from the comparison with observations. Correlation coefficients for the radiation-LWP (*top*) and radiation-$C_V$ regressions (*bottom*) are noted in the top right of each panel.



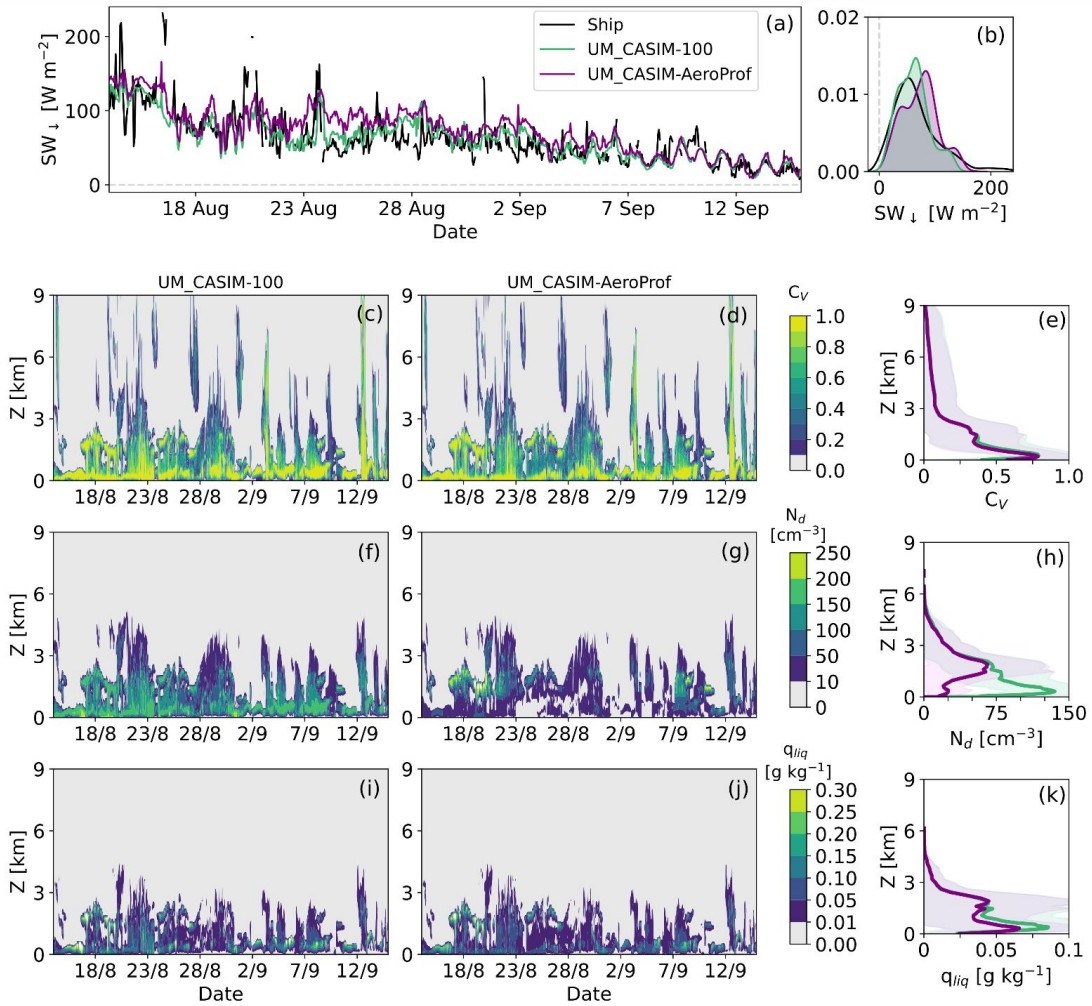

**Figure 7:** Comparison of UM_CASIM-100 and UM_CASIM-AeroProf, demonstrating the influence of representative aerosol concentrations on the modelled cloud structure. (**a—b**) downwelling shortwave radiation ($SW_\downarrow$) at the surface, with observations (black) shown for comparison; (**c—e**) $C_V$; (**f—h**) cloud droplet number concentration ($N_d$); and (**i—k**) liquid water mixing ratio ($q_{liq}$). (**c, f, i**): UM_CASIM-100; (**d, g, j**): UM_CASIM-AeroProf; (**e, h, k**): mean profiles with ± one standard deviation shown in shading. Radiative differences are only notable between 22 Aug and 27 Aug. Slight differences in $q_{liq}$ and cloud fraction can also be identified during this period; for example, UM_CASIM-100 produces a larger cloud fraction below 2 km at 23 Aug



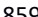

**Figure 8**: $T$ (left) and $q$ (right) measured by the radiosondes over the AO2018 drift period. (**a—b**): radiosonde data re-gridded to the UM vertical grid for model comparisons. (**c—d**): biases of IFS data, re-gridded to the UM vertical grid, with respect to observations. (**e—j**) UM_CASIM-100, UM_RA2T, and UM_RA2M biases, with no vertical re-gridding. The common vertical grid (from the UM) provides 50 vertical levels below 10 km, with 21 of these below 2 km. The black line in all panels depicts the altitude of the main inversion base as identified using the radiosonde measurements, and meteorological time periods with common characteristics are indicated in white (see **Vüllers et al., 2021** for details).


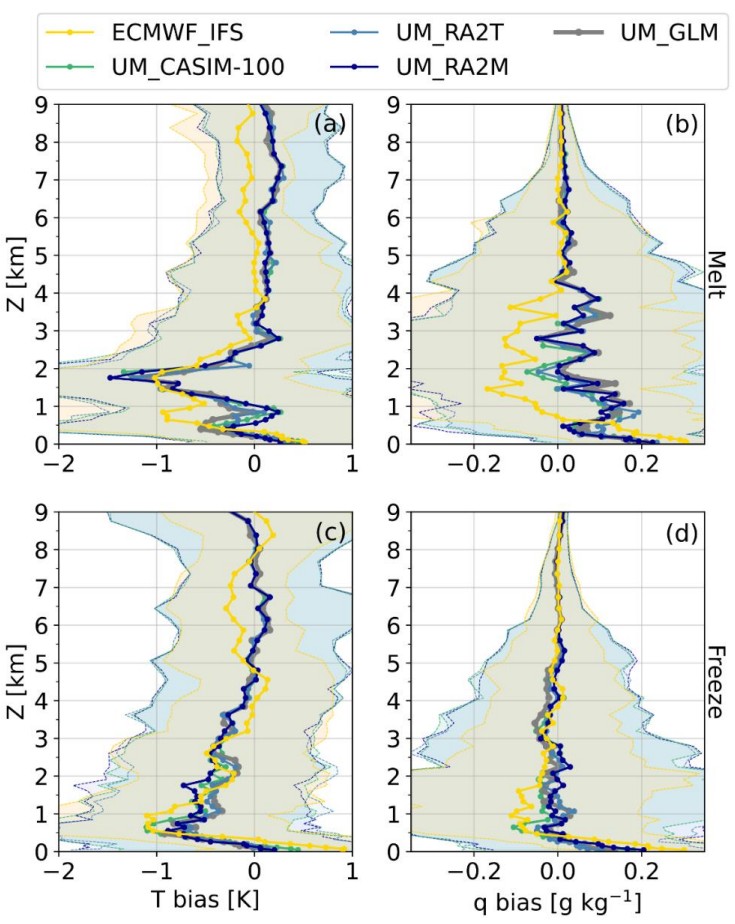

**Figure 9**: Median profiles of modelled (**a, c**) $T$ and (**b, d**) $q$ biases with respect to the radiosonde measurements over the sea ice melt (*top*) and freeze (*bottom*) periods (using 28 Aug as a threshold). Model data are coloured as previous (ECMWF_IFS: yellow; UM_CASIM-100: green; UM_RA2T: light blue; and UM_RA2M: dark blue) and ± one standard deviation shown to illustrate variability. Median anomalies from the UM global model (UM_GLM; grey) are also included for reference; variability is not shown for these data.





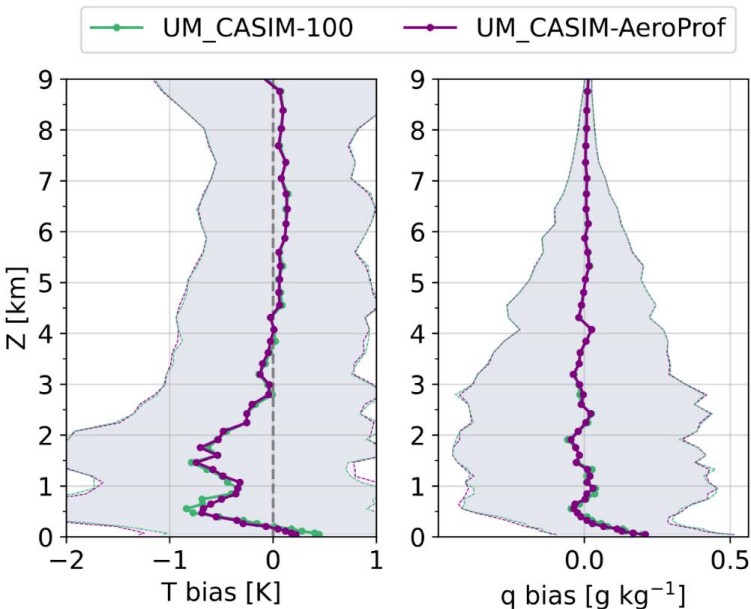

**Figure 10**: Temperature (*left*) and moisture (*right*) biases exhibited by the UM_CASIM-100 (green) and UM_CASIM-AeroProf (purple) simulations with respect to radiosonde measurements made over the entire drift period.  ± one standard deviation shown in shading to illustrate variability.







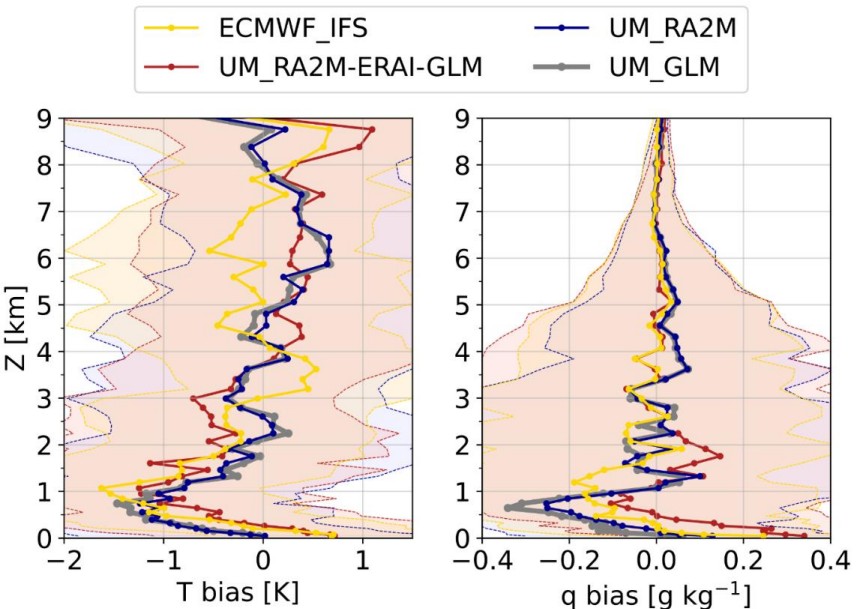

**Figure 11**: Median $T$ and $q$ biases from a subset of the drift (31 Aug to 5 Sep) for ECMWF_IFS (yellow), UM_RA2M-ERAI-GLM (red), UM_RA2M (dark blue), and UM_GLM (grey). UM_RA2M-ERAI-GLM biases follow ECMWF_IFS biases up to approximately 1 km, above which they largely behave more like the other UM cases. ± one standard deviation shown in shading to illustrate variability.






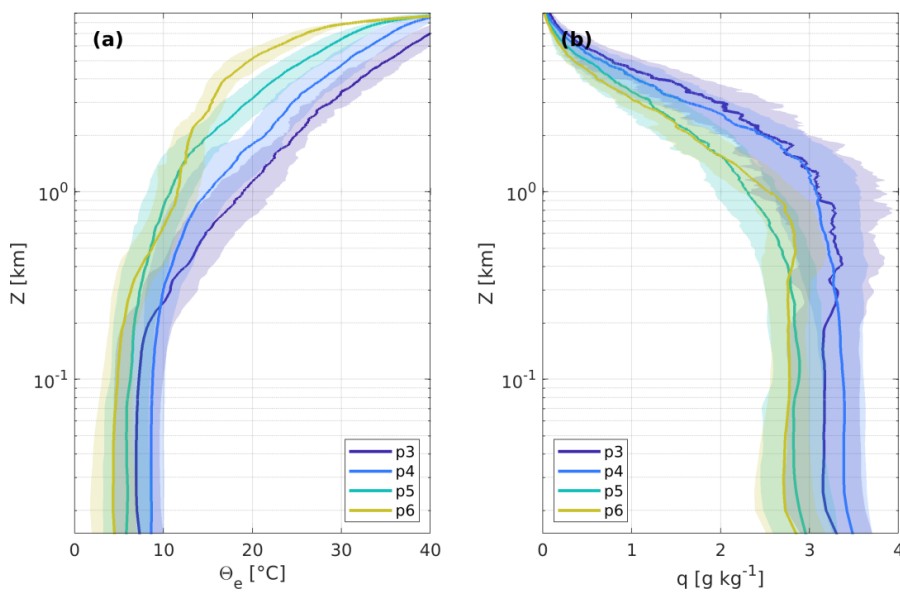

**Figure 12**: Mean profiles of (a) equivalent potential temperature ($\theta_e$) and (b) $q$ measured by radiosondes launched during periods 3—6 of the expedition, with ± one standard deviation shown in shading.


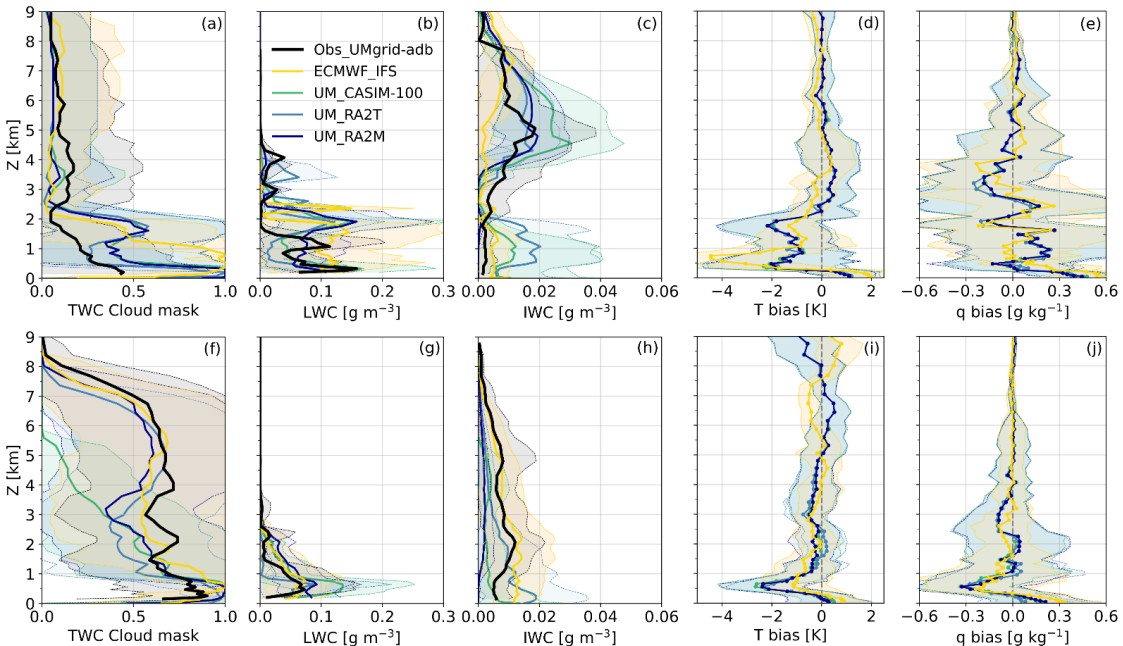

**Figure 13**: Comparison of mean cloud mask, $LWC$, and $IWC$ profiles with median biases in $T$ and $q$ with respect to radiosondes for period 3 (**a—e**, *top row*) and period 6 (**f—j**, *bottom row*). Again, observed $LWC$ calculated assuming adiabatic conditions using Cloudnet. ± one standard deviation shown in shading to illustrate variability.






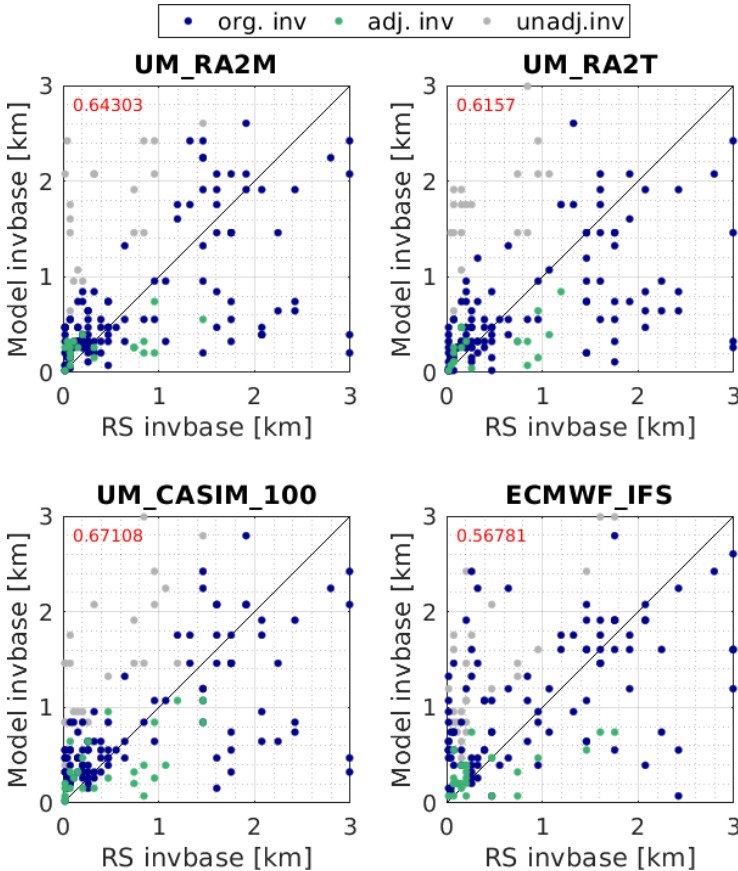

**Figure 14**: Model temperature inversion base as a function of the identified inversion base from radiosonde (RS) measurements. *org. inv*: strongest inversion below 3 km, identified following **Vüllers et al. (2021)**. *adj. inv:* where models exhibit a secondary weaker inversion at lower altitude in better agreement with identified radiosonde inversions, these identified inversions are adjusted accordingly. *unadj.inv*: unadjusted primary inversions, not used for further analysis and shown for reference only. Correlation coefficients are quoted in red in at the top of each panel.






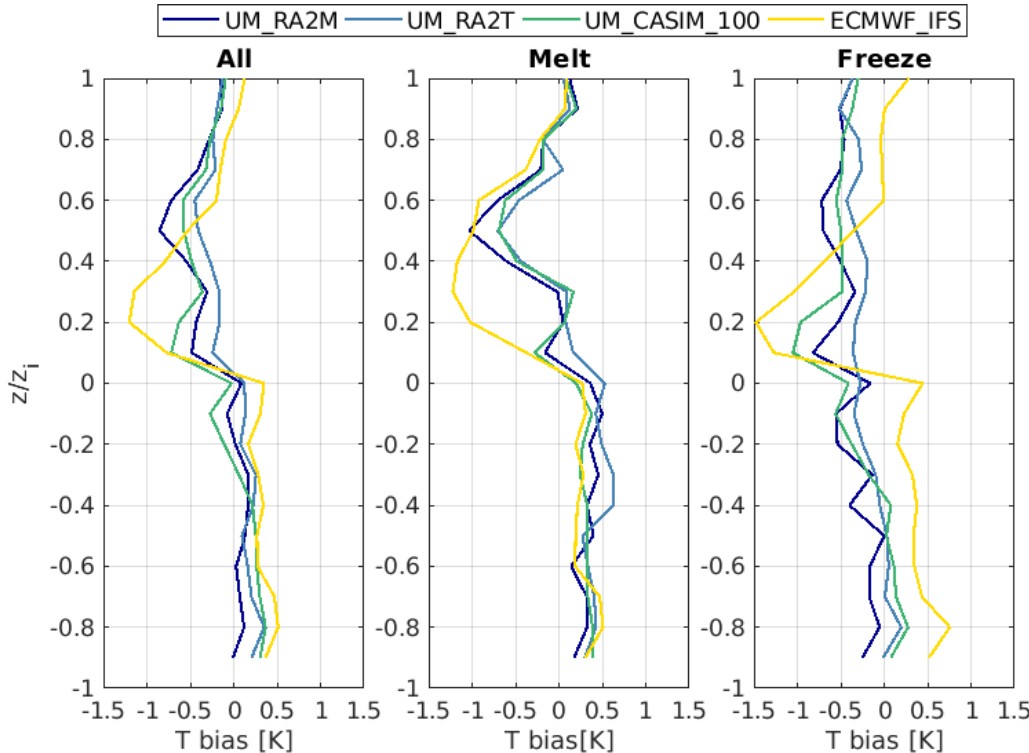

**Figure 15:** Scaled median model-observation $T$ bias profiles for the full drift *(left),* melt *(middle),* and freeze *(right)* periods. Profiles are scaled such that –1 is the surface, 0 is the main inversion base, and 1 is 3 km.






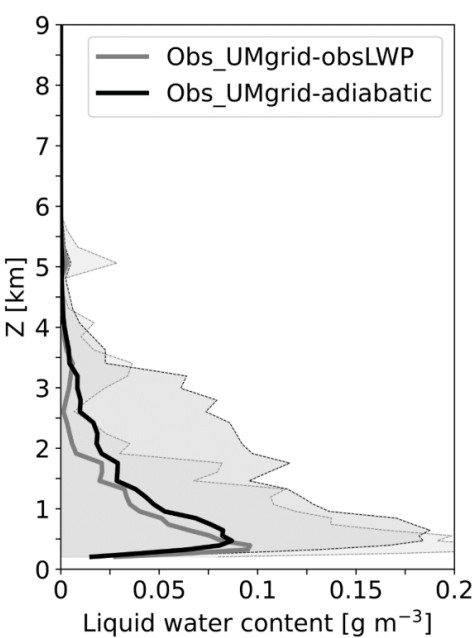

**Figure A1**: Comparison of mean $LWC$ profiles calculated using an adiabatic assumption (black, ± one standard deviation

shown in dark grey shading) and from HATPRO $LWP$ measurements (grey, ± one standard deviation shown in light grey)

without the Cloudnet offsetting procedure.

**Data availability**

UK contributions, as well as selected other data, are available within the MOCCHA (*Microbiology-Ocean-Cloud Coupling in*

*the High Arctic*) data collection in the Centre for Environmental Data Analysis (CEDA) archives (http://archive.ceda.ac.uk/).

Other cruise data are available from the Bolin Centre for Climate Research MOCCHA/AO2018 holdings

(http://bolin.su.se/data).

**Author Contributions**

GY led the model data analysis, aided by PF, RP, JD, and RF. JV led the measurement and Cloudnet data analysis, with

contributions from PA, IMB, MT, and EO. IMB, PA, MT, and JP performed the measurements during AO2018. IMB, MT, JD,

PF, RN, GY and JV all contributed to the study design. All authors contributed to the discussion of results and writing of the

manuscript.

**Competing Interests**

The authors declare that they have no conflict of interest.



## Acknowledgements

This work was supported by the UK Natural Environment Research Council (NERC; grant no. NE/R009686/1) and the Knut and Alice Wallenberg Foundation (grant no. 2016-0024). The Swedish Polar Research Secretariat (SPRS) provided access to the icebreaker Oden and logistical support. We are grateful to the Chief Scientists Caroline Leck and Patricia Matrai for planning and coordination of AO2018, to the SPRS logistical staff and to I/B Oden's Captain Mattias Peterson and his crew.

The Atmospheric Measurements and Observations Facility (AMOF) of the UK National Centre for Atmospheric Science (NCAS) provided the cloud radar, HALO lidar, RPG HATPRO radiometer, Campbell ceilometer, Kipp & Zonen radiometers, and Vaisala radiosounding station. The soundings were supported by Environment and Climate Change Canada in collaboration with the Year of Polar Prediction, Polar Prediction Project.

We acknowledge use of the Monsoon2 system, a collaborative facility supplied under the Joint Weather and Climate Research Programme, a strategic partnership between the Met Office and the Natural Environment Research Council. This work also used JASMIN, the UK collaborative data analysis facility, and was achieved in part with help from the Centre for Environmental Modelling and Computation (CEMAC), University of Leeds.

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
