# Peer review of "Evaluating Arctic clouds modelled with the Unified Model and Integrated Forecasting System"

_Atmospheric Chemistry and Physics, 2021_

## Referee Comment (RC2)

**Review of "Evaluating Arctic clouds modelled with the Unified Model and Integrated Forecasting System" by Young et al.**

The authors analyse the capabilities of the Unified Model and the IFS to accurately capture clouds and surface radiative fluxes measured during the Arctic Ocean 2018 expedition. They present simulations covering roughly a month-long period and they discuss the different performance of the models over this time period. Model simulations differ in various settings and physical parameterisations. While finding systematic biases in cloud cover and to a lesser extend in cloud microphysical structure, thermodynamic and moisture biases inherited from the driving model are found to play a major role.

Overall, the paper presents a very interesting and comprehensive analysis and the paper is generally well-written. The results are mostly presented in a clear manner. However, some clarifications / more extensive presentation are needed in some places, while the paper is repetitive in other sections (details and suggestions in the comments below). However, these are mostly minor issues, that should be easy to fix. If these corrections have been, I recommend the paper to be published.

**Comments**
- l. 96: What hypotheses? The bullet point list given above does not list hypotheses but rather model components to be investigated.
- l. 148-151: Please briefly discuss the spatial heterogeneity or homogeneity of the modelled/ observed cloud fields. Are there any issues to be expected from using the closest grid point to the ship location for verification?
- l. 173 ff: How representative are the climatological values for sea-ice cover and thickness for the month considered here? This could be particular an issue given the rapid shift in conditions due to climate change in recent years.
- l. 178 ff: How are sea-ice cover and sea surface temperature modelled in the UM?
- l. 220 ff: What are the assumptions / settings in the turbulence parameterisation in UM_CASIM-100? Are they similar to UM_RA2M or UM_RA2T or altogether different?
- l. 237 ff and earlier: The claim to test different settings with a simulation where multiple parameters / parameterisations are changed (here: albedo parameterisation and cloud microphysics) is a bit troublesome. Assigning changes or non-changes is difficult if there is more than one scheme changed. The authors should better acknowledge this in the text here and in the presentation of the results later on (btw. this holds also for the comparison of CASIM simulation to UM_RA2M and UM_RA2T).
- l. 255: This is a vital point in the later analysis, therefore a short summary on how periods of "consistent meteorology" are identified and what characterises them is needed here. Only referencing earlier work is not sufficient here.
- l. 304: The differences in the values given here are actually larger than the -1.9 $Wm^{-2}$ maximum bias given above for the simulation claimed to be a worse match to observations.
- l. 347: The failure of the model to reproduce persistent clouds at higher altitudes is actually also notable in the TWC metric.
- l. 403: Give some brief insight in why the ice-phase should be more active in UM_RA2M than UM_RA2T despite them using the same microphysics. Is this due to mixing processes or the differing cloud schemes?
- l. 405 ff: Please clarify if the LWP is calculated for z>150 m in observations and model?
- l. 429 - 432: Please rephrase this sentences and make clearer references to the Figures.
- l. 445: It is fine to have the details of this simulation in the SI, but at least provide the aerosol number concentration in the BL in the main text for comparison with the 100 $cm^{-3}$ simulation.
- l. 490 ff: Would it not make sense to look at relative rather than absolute biases in moisture content. The tampering off above 4 km could be due to the general decrease of q with altitude.
- l. 547 ff: How much of the moisture biases actually translates into an relative humidity bias, which is more important for cloud cover than pure moisture biases.

- l. 606 ff: Can you estimate how much of these biases are actually inherited from the global model? This is discussed later in section 4 and also in the abstract, but initial / lateral boundary condition biases are not mentioned here.
- Section 4: This section essentially is a somewhat more concise presentation of the key results already presented in section 3. This is repeated again in section 5. Hence there is a lot of repetition in these sections, which makes the paper more lengthy and tedious to read than necessary. Please consider either shortening the summary of results in section 4 or moving the few instances of proper discussion / comparison with other studies to section 3 or 5.
- l. 637: Can you at least briefly provide some examples of the "other meteorological factors and incorrect model biases" you mention here.
- l. 679: Despite the present study covering a month of simulations, it is not possible to draw a link between model biases in the present study and climate change signals.

**Technical corrections**
- l. 30: "cloud cover likely contributes to the  near-surface temperature bias"
- l. 153: " from Jun**e** 2019 to Jun**e** 2020"
- l. 338: I do not think surface properties are extensively discussed rather thermodynamic conditions are analysed in the following.
- l. 394: Not sure what you mean with "consistent with altitude" here.
- l. 401/402: What are "shaded standard deviations"?
- l. 416: precipitation is increased compare to what?
- l. 471: replace "noise" with "variability"
- l. 537: "a more prominent than a secondary layer" ? Please rephrase.
- l. 566: Not sure which "effect" you are refering to here.
- l. 728: "less dominant ice **microphysics**"

---

## Author Response (AR1)

The authors would like to extend our thanks to both reviewers – their useful comments and suggestions have enabled us to improve the manuscript.

Reviewer 1 comments shown in red, Reviewer 2 in blue. Authors' response is in black.

Reviewer 1:

This paper compares NWP model output from the UM (using several different cloud schemes) and the IFS against ground-based observations collected in the central Arctic during a summer time cruise that sampled both sea ice melting conditions and refreezing conditions. The results demonstrated that both model frameworks overestimated cloud occurrence, but that using a total water content method, which takes into account the insensitivity of the ground-based remote sensors to very small amounts of cloud water content (or very small particle size) provided better agreement in cloud occurrence. Even still, the models tended to overestimate the LWC relative to the observations, which was hypothesized to result in too much cloud top radiative cooling which had deleterious impacts on the simulated temperature profile. However, they also showed that the UM models, which were all limited area models, sensitive to the forcing conditions from the global model used to drive these simulations. They also showed that a more accurate treatment of aerosols in the UM-LAM with the most complex cloud microphysics did change the profile of LWC in the lowest levels, but had virtually no other effect on the cloud lifetime, precipitation amount, or the biases in the thermodynamic profiles.

I found this paper very interesting, well-motivated, and well written.

My main comment is associated with the 4th point raised in the conclusions:

I think that the sensitivity of the results to the forcing dataset used to drive the UM models casts a lot of questions on this analysis. In section 3.4 (and later sections), the authors work hard to connect errors in clouds to errors in thermodynamic profiles (it seemed like a cause-effect implication). However, I don't think the authors have done enough to convince me that the errors in the clouds are causing the rest of the issues. I think this could be addressed reasonably simply by showing the biases in the thermodynamic profiles over the region from the forcing dataset itself (e.g., in Fig 13 and subsequent figures). I realize that the UM models are providing 12-36 h forecasts that start from the forcing dataset, but I think adding these bias profiles would still be useful in making their case.

- For clarification, we would like to first raise two points in response to the Reviewer's comment:

    1. While we are running T+36h forecasts, lateral boundary conditions for the UM LAM are periodically updated (every hour) from the global model. The global model itself is initialised from archived Met Office analyses at T=0, after which the T+36h global forecast is conducted primarily for the generation of LBCs for the LAM. It was not clear from their comment whether the Reviewer had thought that the LAM was free-running – the global model runs in atmosphere-only mode for 36-h, from which the LBCs for the LAM are generated, so the LAM is not free running.

    2. Given we do not have a good "ground truth" across the high Arctic, with even ERA5 showing thermodynamic biases in the region (see **Tjernström et al., 2021**), we do not have a good observational/reanalysis dataset available for evaluation across a broader region as the reviewer suggests. We have only the point radiosonde launches/observations at the ship, which are already being used for comparison in this study.

       However, following the Reviewer's request, we have now included the thermodynamic biases in the UM global analyses and ERA-Interim in the Supporting Information (Section S4) – we have only included 1200 UTC sonde launches between 31 Aug and 5 Sep to align with the discussion surrounding Figure 11 in Section 3.3.1. Given we have fewer profiles to compare, there is only so much we can take from this comparison. However, a key point to note is that both initial condition datasets lack strong thermodynamic biases in the lower atmosphere; biases that are exhibited by both our LAM and global UM simulations and ECMWF_IFS (Figure 11).

We feel that the discussion surrounding Figure 11 (Section 3.3.1) and new additional analysis of Section 4 now fully addresses the reviewer's concern regarding the forcing dataset. In the UM_RA2M case, LBCs are generated hourly from the UM global model initialised from operational analyses. In the UM_RA2M-ERAI-GLM case, LBCs are generated hourly from the UM global model initialised from ERA-Interim. The only difference between these two simulations is the initial conditions, both of which lack strong thermodynamic biases in the lower atmosphere (excluding the surface; Figure S9). The fact that such low atmosphere biases develop after initialisation, and do so in different ways dependent on the chosen initialisation dataset, is an important finding for the modelling community.

We believe that the excessive cloudiness at low altitude is contributing these biases, but there are other related model issues influencing the thermodynamic profiles. As rightly indicated by the Reviewer, we believe that these biases are somewhat a cause/effect relationship linked with the increased cloudiness in the models. These biases could also manifest from, for example, the modelled BL being too deep at certain timesteps, as a too-high cloud top and inversion height would create a similar local thermodynamic bias as too much cloud top radiative cooling/cloudiness. Similarly, strong, sharp biases could result from the models failing to capture an inversion that was observed, and this happened frequently. However, while both of these scenarios were identified on occasion, Figure 15 indicates that a misplaced boundary layer inversion doesn't explain the strong cold biases at cloud height alone, and the cloud top radiative cooling itself appears to be driving this.

The thermodynamic biases aloft (above approximately 2 km) do not appear to be tied so clearly to cloud. The UM analyses exhibit increasing temperature biases with height and ERA-Interim exhibits a cold bias between 2 km and 8 km, both of which are concerning and somewhat explain the similar behaviour of the UM and IFS thermodynamic biases aloft (though, strictly speaking, ECMWF_IFS is not initialised from ERA-Interim). These biases therefore are less clearly tied to errors in the modelled clouds, and likely have a different source.

In a very similar and related comment: it looks like the biases in the LWC profiles from the three UM models are quite different, but the biases in the temperature and moisture profiles are essentially identical. If errors in the cloud properties are truly the driver (via radiation) of the biases in the temperature profiles, then I would have hypothesized we would see differences in the biases in the temperature profiles from the three models. Why don't we?

- We appreciate that these differences were very subtle in the original version of the manuscript, so we have added some additional information to the revised manuscript (Section 3.3) in addition to an extra section (Section 4) in the Supporting Information with more details on how the UM LAM biases differ from each other and from the UM global model. For three example 36-h forecasts, we have included the biases of all UM simulations with respect to the radiosondes, showing that each of the LAM simulations have a bias structure which does align suspiciously well with the global model biases. However, we subsequently show that, on an individual profile (6-hourly) basis within an example 36-h forecast, the LAM simulations do indeed exhibit biases in temperature, moisture, and cloud liquid water mixing ratio with respect to the global model.

However, from the original manuscript, we would argue that we do already see these differences, though we agree that they were perhaps not emphasised clearly enough.

Take Figures 7 and 10 for example (the comparison between UM_CASIM-100 and UM_CASIM-AeroProf). These simulations differ little in model-diagnosed $C_v$ but differ significantly in $N_d$ (particularly around 500 m, where UM_CASIM-100 peaks in $N_d$ whereas UM_CASIM-AeroProf has significantly fewer cloud droplets). This is reflected in a reduced $q_{liq}$ between approximately 500m and 1.5 km in UM_CASIM-AeroProf, with comparison to UM_CASIM-100.

This subtle difference is reflected in the thermodynamic biases exhibited by these simulations – UM_CASIM-100 has a stronger negative temperature bias at 500 m than UM_CASIM-AeroProf, and a warmer BL towards the surface, likely caused by the warming effect from an overestimated cloud $LWC$ and amount of cloud cover (as noted in Reviewer 1's point below).

Minor comments:

- The differences between the model diagnosed cloud cover and the cloud cover derived from the TWC "cloudNet simulator" was striking. I feel that there was too much emphasis on the Cv estimate; I believe that we need to use instrument simulators much more routinely in model – observation comparisons. I would like to see this emphasized more in the conclusions.

  - We agree that this difference is important and should be highlighted; however, we would disagree that the emphasis was on the $C_V$ estimate in the original manuscript. The difference between the described cloud fraction metrics was included in Figures 3 and 4 to demonstrate the clear difference between our $TWC$-derived cloud mask and $C_V$/model-diagnosed cloud fraction, but the former was used predominantly throughout our analysis.

    We used $C_V$ in Figures 6 and 7 for good reasons: Figure 6 because we required a metric independent of $LWC$ to evaluate correlations, and Figure 7 to clearly demonstrate the issue with using the Smith (1990) large-scale cloud scheme (with only subtle differences in $q_{liq}$, the $C_V$ was largely unchanged due to the simple relationship between these parameters).

    We have added an extension to our first conclusion point to make it clear that we are advocating the use of the $TWC$ cloud mask for instrument comparisons rather than the model-diagnosed cloud fraction:

    *"... and we would advocate for the use of cloud water contents to derive comparable cloud occurrence metrics between observations and models."*

- Line 389: that all model simulations overestimate LWC in the 1-3 km range relative to the observations is interesting, especially since the obs are using an adiabatic assumption to distribute the liquid water. Thus, the true bias in LWC in the models is likely even larger than what was shown. I think this should be pointed out somewhere in the paper.

  - We agree with Reviewer 1 and have added the following lines to Section 3.2, where the use of the adiabatic assumption is introduced and in the initial comparison with models:

    *"However, we must note that this assumption likely overestimates the observed $LWC$ as these clouds are likely sub-adiabatic."*

    *"… Considering that we employ an adiabatic assumption for our observations, thereby giving an upper limit for the observed $LWC$, these model $LWC$ biases are likely greater in reality than shown here."*

- Fig 6: the units of LWP are incorrect; I suspect they should be g/m2

  - That is correct – thank you for highlighting this mistake, it has been rectified in the updated manuscript.

- Lines 472-474: the water vapor units are g/kg, not g/m3

  - Fixed in revised manuscript

- The yellow color used to denote the IFS results is too faint to see well; please increase its contrast

  - We chose these colours to ensure our figures were accessible and readable by colourblind viewers; however, we do appreciate that the yellow lines denoting the IFS data were difficult to see in some of the figures. We have increased contrast in all figures by darkening the yellow colour and, where the yellow line was immediately on top of the white background, we have added a slight grey background (e.g., Figures 2, 5, 6 and 15).

- Line 522: Satellites provide good coverage of the arctic, and the infrared sounders do provide thermodynamic profiles (of some quality, depending on your metric). I think that some mention to the challenge of using these satellite data for DA is needed here.

- We understand the reviewers point and appreciate that satellite data are valuable for the Arctic; however, recent studies have shown that these data do not compensate for in-situ data coverage from radiosoundings when it comes to improving biases in analyses. See results of e.g., Naaka et al., 2019 (GRL). We have added the following sentence to illustrate this issue to the reader:

  "*The comparatively comprehensive spatial coverage from satellites does not compensate for good in-situ observations from radiosoundings and does little to correct a biased model DA input (**Naaka et al., 2019**).*"

- LW radiative cooling is strongly dependent on (a) the integrated water content of the cloud and (b) if there is another cloud above the radiating layer or not. Turner et al. JAMC 2018 provides a good illustration of this for arctic clouds. Line 562 is hypothesizing too much LW radiative cooling, but we have seen that the different microphysics parameterizations yield different LWCs. Is this because there are clouds above this BL that is muting this radiative impact somehow?

  - This could likely be a contributing factor to why the biases are not necessarily consistent with the cloud microphysical structure alone – the cloud scene was more often multi-layered rather than single-layered. Where we have single-layered clouds, during Period 3 for example, the biases are far stronger than during periods with many multi-layered clouds (e.g., Period 6; Fig. 13). We have added the following to Section 4.2.1:

    "*Similarly, **Turner et al. (2018)** note that the presence of cloud aloft can significantly modulate the radiative cooling response of low-level Arctic clouds – in Period 6, multi-layered clouds were prevalent, thereby potentially muting the radiative impact. In Period 3, however, few clouds were observed and those which were present often occurred in single layers (**Vüllers et al., 2021**); during this period, we found the greatest thermodynamic biases in our models with respect to our observations.*"

- Line 591: are the correlation coefs on Fig 14 for the "orig inv" or the "adj inv" dataset? It is not clear

  - These correlation coefficients relate to the combined *org. inv* plus *adj. inv* dataset, i.e., all data excluding the unadjusted inversion heights, shown only for reference. This information has been added to the caption of Figure 14.

- Line 607-609: if there is too much radiative cooling at cloud top (because the LWC is too high), then this would result in greater LW radiative warming in the lower part of the BL, which could lead to this warm bias in the surface (a possible explanation for the warm bias near the surface).

  - We thank the reviewer for their insight – we completely agree and do think that $LW$ warming in the lower BL is driven as they suggest. We suspect it is a combination of this and an issue with the too-warm surface (as has been reported previously). We have added the following to the revised manuscript (in Sect. 4.1.2: Longwave to link with the existing discussion) to reflect the reviewer's input:

    "*The overestimation of cloud cover and $LWC$, which drives too much radiative cooling at cloud top, will also result in an excess downward $LW$ flux which would act to warm the lower BL and thus contribute this warm bias, yet it is not clear whether the too-warm surface (as reported previously by **Birch et al., 2009; Sotiropoulou et al., 2016**) is similarly a result of incorrectly modelled cloud or whether it is a separate issue*"

    This sentence is placed before the inclusion of evidence from **Tjernström et al. (2021)**, which suggests that the atmosphere is indeed warming the surface.

Reviewer 2:

The authors analyse the capabilities of the Unified Model and the IFS to accurately capture clouds and surface radiative fluxes measured during the Arctic Ocean 2018 expedition. They present simulations covering roughly a month-long period and they discuss the different performance of the models over this time period. Model simulations differ in various settings and physical parameterisations. While finding systematic biases in cloud cover and to a lesser extend in cloud microphysical structure, thermodynamic and moisture biases inherited from the driving model are found to play a major role.

Overall, the paper presents a very interesting and comprehensive analysis and the paper is generally well-written. The results are mostly presented in a clear manner. However, some clarifications / more extensive presentation are needed in some places, while the paper is repetitive in other sections (details and suggestions in the comments below). However, these are mostly minor issues, that should be easy to fix. If these corrections have been, I recommend the paper to be published.

**Comments** –
- l. 96: What hypotheses? The bullet point list given above does not list hypotheses but rather model components to be investigated.

    o We have changed this to "*testing these model components*"

- l. 148-151: Please briefly discuss the spatial heterogeneity or homogeneity of the modelled/ observed cloud fields. Are there any issues to be expected from using the closest grid point to the ship location for verification?

    o There are, of course, differences expected when comparing an average over a region to a point-by-point extraction of variables from a 4D dataset. We have included a discussion on the spatial heterogeneity of the key cloud fields included in our discussion (Cloud fraction, liquid water mixing ratio, and ice mixing ratio) in the Supporting Information (Section S3) – to summarise, cloud fraction and $q_{liq}$ vary only a little between an average over an area, or "swath", and the point comparison included in the main body of this study.

    There is more variability in the $q_{ice}$ fields, particularly during the freeze up period; however, this is to be expected due to the prevalence of scenarios with either very low or much higher ice contents (in the colder clouds). The ice phase has not been our primary focus in this study, yet we do note that it is highly variable; sampling over more data logically improves the statistics.

    A timeseries of cloud variables is required for input to Cloudnet; it is designed to be used from a single site, and so is not equipped to be run across a region. Our comparison of the model variables (now included in the Supporting Information) shows that the model fields that are key to our discussion, particularly cloud fraction and $q_{liq}$, do not vary substantially if a swath average is taken; therefore, we justify the use our point comparison of variables extracted from the ship's location solely within the Cloudnet framework here.

- l. 173 ff: How representative are the climatological values for sea-ice cover and thickness for the month considered here? This could be particular an issue given the rapid shift in conditions due to climate change in recent years.

    o We do not directly address the sea ice areal coverage or thickness in this study as the expedition was making measurements very close to the North Pole. The sea ice in this area has not yet seen significant summertime melt and so ice cover should be close to the climatology in this region. If we were measuring closer to the ice edge, we agree that an assessment of sea ice modelling capability with respect to the climatology would be an important factor.

- l. 178 ff: How are sea-ice cover and sea surface temperature modelled in the UM?

  - Sea surface temperature and sea ice concentration are assimilated from daily OSTIA products into the operational analyses used as initial conditions for the global model. These conditions are fixed for the short duration of each 36h forecast, though they are suitably updated at the initialisation of each subsequent forecast.

- l. 220 ff: What are the assumptions / settings in the turbulence parameterisation in UM_CASIM-100? Are they similar to UM_RA2M or UM_RA2T or altogether different?

  - Representation of turbulence is the same in UM_CASIM-100, UM_RA2M, and UM_RA2T; the only changes are in the large-scale cloud, microphysics, and sea ice albedo schemes as described in the manuscript. In these UM simulations, mixing in the vertical is driven by a non-local 1D boundary layer scheme described in Lock et al. (2000). However, one must note that interactions are probable between the turbulence and microphysics schemes via radiatively-driven cooling at cloud top and resultant convective overturning which is responsive to changes in the cloud microphysical structure. We have added the following to Sect. 2.3.2 of the manuscript:

    *"Each simulation uses the same boundary layer scheme, where mixing in the vertical is described by Lock et al. (2000); however, one must note that turbulent interactions can be influenced by the relationship between cloud top radiative cooling and subsequent convective overturning with cloud microphysics."*

- l. 237 ff and earlier: The claim to test different settings with a simulation where multiple parameters / parameterisations are changed (here: albedo parameterisation and cloud microphysics) is a bit troublesome. Assigning changes or non-changes is difficult if there is more than one scheme changed. The authors should better acknowledge this in the text here and in the presentation of the results later on (btw. this holds also for the comparison of CASIM simulation to UM_RA2M and UM_RA2T).

  - We appreciate that changing both the microphysics and albedo parameterisation is unusual for a methodical approach to model evaluation, but we felt it important to include the updated albedo case in this study as it demonstrates some interesting results. We did conduct a CASIM simulation of just changing the microphysics and keeping the sea ice albedo the same as in the UM_RA2M and UM_RA2T cases – results from this simulation are shown in the Supporting Information (Section S2) and not included in the main manuscript to avoid overcrowding the figures. This CASIM simulation (with the same albedo settings as UM_RA2M and UM_RA2T) is not significantly different from UM_CASIM-100 in terms of the modelled cloud amount and microphysical structure; the key difference is seen in the $SW_{net}$. We have included mean $LWC$ and $IWC$ comparisons (similar to that shown in Figure 4) in the Supporting Information to illustrate the minor difference in cloud properties caused by the change in albedo settings alone (now Figure S6).

- l. 255: This is a vital point in the later analysis, therefore a short summary on how periods of "consistent meteorology" are identified and what characterises them is needed here. Only referencing earlier work is not sufficient here.

  - We have added the following to the revised manuscript to reflect the metrics by which consistent meteorology was determined:

    "*... based on similarity of equivalent potential temperature and relative humidity profiles measured, as defined in* **Vüllers et al., (2021)**"

- l. 304: The differences in the values given here are actually larger than the -1.9 Wm-2 maximum bias given above for the simulation claimed to be a worse match to observations.

  - We take the Reviewer's point and understand that we may have caused confusion here – when stating that agreement was improved, we were referring to the shape of each

of the PDFs rather than the median value. This has been made clearer in the revised manuscript.

- l. 347: The failure of the model to reproduce persistent clouds at higher altitudes is actually also notable in the TWC metric.

  - We agree with Reviewer 2 – we have separated the $C_V/TWC$ discussion here to make clear how each metric compares with the observations before comparing their performance, as this combined discussion was making this key point less clear to the reader. We have made it clearer in the revised manuscript that we can also see this underestimation of cloud aloft from the $TWC$ metric later in this section when it is discussed:

    "*Above 2 km, each model simulation underestimates the observed cloud occurrence, in line with the $C_V$ metric comparison.*"

- l. 403: Give some brief insight in why the ice-phase should be more active in UM_RA2M than UM_RA2T despite them using the same microphysics. Is this due to mixing processes or the differing cloud schemes?

  - Mixing processes are unchanged between these two simulations, but UM_RA2M and UM_RA2T use different large-scale cloud schemes – the latter, UM_RA2T, uses the prognostic cloud condensate (PC2) scheme which represents a prognostic ice mass mixing ratio, from which a bespoke ice cloud fraction is calculated, and includes a distribution of $IWC$ for the same cloud fraction. In the Smith (1990) scheme used in UM_RA2M, $IWC$ is fixed for a given temperature and only the total cloud fraction is represented. This is discussed in the Supporting Information, Sect. S1.1; however, we have summarised this information for clarity in Sect. 2.3.2.1 of the manuscript.

- l. 405 ff: Please clarify if the LWP is calculated for z>150 m in observations and model?

  - LWP is measured by the RPG HATPRO microwave radiometer, and so is not subject to the height limitations introduced by the radar. All altitudes are included in the LWP comparison shown in the manuscript (Fig. 5).

- l. 429 - 432: Please rephrase this sentences and make clearer references to the Figures.

  - We have rephrased this segment of text as follows:

    "*Positive $C_V$ biases appear to coincide with positive $LWP$ biases, negative $SW_\downarrow$ biases (**Fig. 6a, d, g, j**), and positive $LW_\downarrow$ biases (**Fig. 6b, e, h, k**), and vice versa, indicating that too much cloud cover, and too much cloud liquid water, is tied to the radiative biases shown. The correlation with $LWP$ bias is weaker for $R_{net}$ than for $SW_\downarrow$ or $LW_\downarrow$, likely due to the additional influence of other factors (e.g., surface albedo) on the net radiative properties.*"

- l. 445: It is fine to have the details of this simulation in the SI, but at least provide the aerosol number concentration in the BL in the main text for comparison with the 100 cm-3 simulation.

  - This information has been added to Section 3.2.1 as requested:

    "*UM_CASIM-AeroProf has a mean $N_d$ of 20.9 ± 15.9 cm$^{-3}$ below 500 m, compared with 101.0 ± 40.2 cm$^{-3}$ in this altitude range for UM_CASIM-100... UM_CASIM-AeroProf uses mean accumulation mode number concentration of 18.5 ± 11.4 cm$^{-3}$ below 500 m which, with comparison to the 100 cm$^{-3}$ specified for UM_CASIM-100, is more appropriate for the region.*"

- l. 490 ff: Would it not make sense to look at relative rather than absolute biases in moisture content. The tampering off above 4 km could be due to the general decrease of q with altitude.

  - We are interested in the lower troposphere (< 4 km) biases and how they relate to low-level cloud in this study. The Reviewer is correct that biases higher in the troposphere would be clearer to visualise if we use a percentage, rather than absolute, bias in moisture. However, we would use such a relative metric with caution at altitudes towards 9 km – the specific humidity itself is very low, and so small differences would manifest as large percentage biases.

    At altitude, the uncertainty associated with the specific humidity measurement from the radiosondes and the model uncertainties in this metric will effectively mask the model-observation biases. Therefore, with a moisture metric expressed as a percentage, these biases would appear to be statistically significant, when in fact they are not. This would leave such a visualisation open to misinterpretation.

    However, given the upper troposphere is not our focus, we have included absolute moisture biases, with respect to the campaign radiosondes, from the model simulations.

- l. 547 ff: How much of the moisture biases actually translates into a relative humidity bias, which is more important for cloud cover than pure moisture biases.

  - It makes sense to approach the biases from the point of view of specific humidity as this is a prognostic variable in each of the model simulations. Relative humidity would require a calculation involving temperature, which we have shown also exhibits biases with respect to observations. Therefore, for a more robust metric for humidity bias, we have chosen to show specific humidity.

    **Tjernström et al. (2021; *QJRMS*)** note that the errors for temperature and humidity compensate to produce a <±3% error in relative humidity (for work with the IFS operational analyses comparing also with measurements from this campaign). This information has been added to the first paragraph of Sect. 3.3 in the updated manuscript.

    "*Specific humidity is considered here as a relative humidity comparison would require a calculation involving temperature, which also shows some biases: Tjernström et al. (2021) note that the errors for temperature and humidity compensate to produce a <±3% error in relative humidity (for work with the IFS operational analyses comparing also with measurements from this campaign). This error was found to be positive below 1 km and negative around 3–5 km, and the magnitude of these errors are within the measurement accuracy.*"

- l. 606 ff: Can you estimate how much of these biases are actually inherited from the global model? This is discussed later in section 4 and also in the abstract, but initial / lateral boundary condition biases are not mentioned here.

  - This is a good point and unfortunately difficult to answer. Reviewer 1 had a similar concern regarding the initial conditions, and we have addressed this above, though we will reiterate here for Reviewer 2:

    Lateral boundary conditions for the LAMs are generated at hourly intervals from the global model and, given the frequency of our radiosonde launches (6-hourly), this gives us two issues when trying to disentangle our biases. Firstly, we do not have the measurement data available to accurately compare with the LAMs at intervals more frequent than the LBC generation or sonde launches. Secondly, we do not have an adequate amount of LAM "free running" simulation time to assess how the LAM biases change temporally at times far from the LBC regeneration.

We have included further analysis of the LAM biases with respect to the global model in the Supporting Information (Sect. S4), from a forecast-by-forecast perspective. Three example forecasts were chosen which span weak and strong thermodynamic biases with respect to the radiosondes. Figure S10 shows that the LAM and global simulations do appear to have very consistent biases with respect to the radiosondes, yet Fig. S11 subsequently shows that that the LAMs do in fact exhibit biases with respect to the global model too; the global and LAM biases are similar, but not identical. Each of the LAM simulations – UM_CASIM-100, UM_RA2M, and UM_RA2T – have different biases, linking with their different representations of cloud.

We believe that the temperature/moisture biases exhibited by the UM LAM simulations at low altitude are strongly affected by cloud; however, we do not believe that this is the only contributing factor. Excess low-altitude cloudiness does not explain thermodynamic biases aloft, for example. Inclusion of the biases exhibited by the initial condition datasets compared in Sect. 3.3.1 show that the biases aloft are likely also inherited by the LAM from the global model. The UM analyses exhibit increasing temperature biases with height and ERA-Interim exhibits a cold bias between 2 km and 8 km, both of which somewhat explain the similar behaviour of the UM and IFS thermodynamic biases aloft (though, strictly speaking, ECMWF_IFS is not initialised from ERA-Interim).

- Section 4: This section essentially is a somewhat more concise presentation of the key results already presented in section 3. This is repeated again in section 5. Hence there is a lot of repetition in these sections, which makes the paper more lengthy and tedious to read than necessary. Please consider either shortening the summary of results in section 4 or moving the few instances of proper discussion / comparison with other studies to section 3 or 5.

  o Section 4 puts the key results in context with other studies and offers potential explanations for some of the differences we see between the models and observations. We have shortened the Conclusions to contain only key information and reduce the repetition from Section 4. We have specifically focused on removing comparisons with other studies in Section 5 and included this discussion in Section 4 only.

- l. 637: Can you at least briefly provide some examples of the "other meteorological factors and incorrect model biases" you mention here.

  o This suggestion was included to demonstrate that we by no means think that we have solved, or even identified, all the model problems. For example, **Tjernström et al. (2021)** show that mean sea level pressure anomalies clearly develop with time through meteorological forecasts – an aspect we do not consider in this study.

  As requested by the Reviewer, we have included examples in the text to indicate where there are other problems in the models beyond the scope of those examined in this study:

  *"…however, it is highly likely that other meteorological factors (e.g., mean sea level pressure anomalies, and subsequent influence on cloud dynamics) and incorrect model processes (e.g., turbulent flux biases) are contributing to this warm surface bias across all our simulations, in addition to cloudiness (Tjernström et al., 2021)."*

- l. 679: Despite the present study covering a month of simulations, it is not possible to draw a link between model biases in the present study and climate change signals.

  o We do not draw such a link; we note that the referenced climate studies have found that increased poleward moisture transport may promote more cloud formation in the central Arctic. This statement is included to demonstrate that this is not necessarily a new finding and could be in line with existing literature.

**Technical corrections**
- l. 30: "cloud cover likely contributes to the ever-present near-surface temperature bias"

Updated (removed "*simulated*" after this statement) in the revised manuscript.

- l. 153: " from Jun**e** 2019 to Jun**e** 2020"

  Updated in the revised manuscript.

- l. 338: I do not think surface properties are extensively discussed rather thermodynamic conditions are analysed in the following.

  Reference to surface properties removed.

- l. 394: Not sure what you mean with "consistent with altitude" here.

  The mean $LWC$ from the ECMWF_IFS does not vary much between approximately 0.5 and 2 km – we have rephrased this statement for clarity.

- l. 401/402: What are "shaded standard deviations"?

  The variability in each dataset is visualised through shaded regions each side of the mean, specifically ± one standard deviation is shown. The further this shaded region extends from the mean, the more variability there is within the dataset. Additional text has been included in the manuscript to help the reader:

  "*Shaded areas, depicting ± one standard deviation from the mean, also indicate…*"

- l. 416: precipitation is increased compare to what?

  We refer to times where the precipitation was measured to increase with respect to the low background level, as the clouds observed (and modelled) were only weakly precipitating for the majority of the time:

  "*… capture the short episodes where more precipitation was observed*"

- l. 471: replace "noise" with "variability"

  Updated as suggested in the revised manuscript.

- l. 537: "a more prominent than a secondary layer" ? Please rephrase.

  We apologise for the confusion; this text was left from an earlier draft. We have updated this statement to read:

  "*... a more prominent secondary layer at 1.6 km*"

- l. 566: Not sure which "effect" you are refering to here.

  We refer to the apparent relationship of moisture biases with a strong temperature bias, as seen in our IFS simulation. The UM, on the other hand, does not demonstrate this same behaviour. We have updated the text for clarity:

  "*... the similarly strong cold bias during period 6*"

- l. 728: "less dominant ice **microphysics**"

  Updated in revised manuscript.